# Lenalidomide derivatives and proteolysis-targeting chimeras for controlling neosubstrate degradation

Satoshi Yamanaka [1,2], Hirotake Furihata[1,3], Yuta Yanagihara[4], Akihito Taya[5], Takato Nagasaka[5], Mai Usui[5], Koya Nagaoka[1], Yuki Shoya[1], Kohei Nishino[6], Shuhei Yoshida[4], Hidetaka Kosako [6], Masaru Tanokura [3], Takuya Miyakawa [3,7], Yuuki Imai [4], Norio Shibata [5] & Tatsuya Sawasaki [1]✉

Lenalidomide, an immunomodulatory drug (IMiD), is commonly used as a first-line therapy in many haematological cancers, such as multiple myeloma (MM) and 5q myelodysplastic syndromes (5q MDS), and it functions as a molecular glue for the protein degradation of neosubstrates by CRL4[CRBN]. Proteolysis-targeting chimeras (PROTACs) using IMiDs with a target protein binder also induce the degradation of target proteins. The targeted protein degradation (TPD) of neosubstrates is crucial for IMiD therapy. However, current IMiDs and IMiD-based PROTACs also break down neosubstrates involved in embryonic development and disease progression. Here, we show that 6-position modifications of lenalidomide are essential for controlling neosubstrate selectivity; 6-fluoro lenalidomide induced the selective degradation of IKZF1, IKZF3, and CK1α, which are involved in anti-haematological cancer activity, and showed stronger anti-proliferative effects on MM and 5q MDS cell lines than lenalidomide. PROTACs using these lenalidomide derivatives for BET proteins induce the selective degradation of BET proteins with the same neosubstrate selectivity. PROTACs also exert anti-proliferative effects in all examined cell lines. Thus, 6-position-modified lenalidomide is a key molecule for selective TPD using thalidomide derivatives and PROTACs.

Lenalidomide, a thalidomide derivative, is an immunomodulatory drug (IMiD) widely used for treating several haematological cancers[1–3], such as multiple myeloma (MM) and 5q myelodysplastic syndromes (5q MDS). IMiDs are a class of drugs that function as a molecular glue and induce 26S proteasomal degradation of neosubstrates by hijacking E3 ubiquitin ligase CRL4[CRBN] via interactions between IMiDs and cereblon (CRBN)[4–7]. In the last decade, many neosubstrates involved in

the molecular action of IMiDs have been reported[5–10]. For example, degradation of IKZF1 and IKZF3 or IKZF1 and CK1α is involved in anti-MM[5,6] or anti-5q MDS activity[7], respectively. However, SALL4 and PLZF degradation is considered the cause of thalidomide teratogenicity[8–10], although this has not been proven in mammalian models.

Targeted protein degradation (TPD) is a powerful mechanism of action of drugs and a fundamental approach to developing drugs for

[1]Division of Cell-Free Sciences, Proteo-Science Center, Ehime University, Matsuyama 790-8577, Japan. [2]Division of Proteo-Interactome, Proteo-Science Center, Ehime University, Matsuyama 790-8577, Japan. [3]Department of Applied Biological Chemistry, Graduate School of Agricultural and Life Sciences, The University of Tokyo, Tokyo 113-8657, Japan. [4]Division of Integrative Pathophysiology, Proteo-Science Center, Ehime University, Toon 791-0295, Japan. [5]Department of Life Science and Applied Chemistry, Nagoya Institute of Technology, Nagoya 466-8555, Japan. [6]Division of Cell Signaling, Fujii Memorial Institute of Medical Sciences, Tokushima University, Tokushima 770-8503, Japan. [7]Graduate School of Biostudies, Kyoto University, Kyoto 606-8502, Japan. ✉e-mail: sawasaki@ehime-u.ac.jp

undruggable proteins[11,12], including transcription factors. Many next-generation thalidomide derivatives, such as CC-90009[13] and CC-122[14], are actively being developed. However, thalidomide derivatives have not been reported to induce selective degradation of therapeutic target proteins without degrading neosubstrates involved in teratogenicity.

Proteolysis-targeting chimera (PROTAC) protein degraders are an alternative approach for TPD[15]. PROTACs are synthesised from two functional compounds: E3 ligase binder-like IMiDs and a target binder such as a target protein inhibitor[16,17]. PROTACs can theoretically target many proteins by using diverse target binders[18–21]. Owing to the remarkable clinical success of IMiDs, TPD is a promising approach for treating several diseases, and many PROTACs[19–21] are being developed worldwide. Several PROTACs have been evaluated in clinical trials[22]. Many E3 binders have been developed, including the CRBN binder[16,23], von Hippel–Lindau (VHL) binder[23,24], cellular inhibitor of apoptosis protein (cIAP) binder[23,25], mouse double minute 2 homolog (MDM2) binder[23,26], DDB1 and CUL4 associated factor 16 (DCAF16) binder[23,27], and DCAF11 binder[28]. Because the CRBN binder has the smallest molecular weight among frequently used E3 binders such as VHL and IAP ligands, and CRBN is ubiquitously expressed in diverse tissues, IMiD-based PROTACs can be applied to PROTACs for diverse target proteins[19,20]. However, IMiD-based PROTACs induce the protein degradation of neosubstrates, such as IKZF1, SALL4, and PLZF[8,20,29,30].

Many structural studies have shown that IMiDs bind to a hydrophobic pocket in the C-terminal domain of CRBN[31,32]. Thalidomide and thalidomide derivatives have two chemical rings: glutarimide and phthalimide rings. The former binds to the C-terminal region of CRBN[31,32], and the latter leads to selective interactions with neosubstrates[33–35]. The phthalimide ring differs between thalidomide and thalidomide derivatives; thus, the chemical functional groups in this ring are crucial to the selectivity towards the neosubstrate[33–35]. In addition, 5-hydroxylation of the phthalimide ring alters neosubstrate selectivity, resulting in 5-hydroxythalidomide, which strongly degrades SALL4 but not IKZF1[10,35]. This evidence leads us to the simple hypothesis that sophisticated chemical modulation of the phthalimide ring in thalidomide derivatives could tightly control the selectivity towards the neosubstrate by constructing a CRBN binder for highly selective TPD.

Here, we show that 6-position modification of lenalidomide is a viable approach for enhancing selectivity of neosubstrates involved in the anti-haematological cancer effect over neosubstrates involved in teratogenicity. 6-fluoro lenalidomide more strongly degraded IKZF1, IKZF3, and CK1α than lenalidomide, showing a high anti-proliferative effect on MM- and 5q MDS-derived cell lines. In contrast, 6-fluoro lenalidomide degraded SALL4 and CK1α less than lenalidomide. PROTACs based on 6-modified lenalidomide for BET proteins have the same neosubstrate selectivity as 6-modified lenalidomides and anti-proliferative effect on all examined cell lines. Therefore, our data show that 6-position modification of lenalidomide is a viable approach for selective TPD by thalidomide derivatives and PROTACs.

## Results

### 6-position modifications on lenalidomide
To identify thalidomide derivatives that selectively induce protein degradation of therapeutic targets for haematological cancer, we synthesised 10 thalidomide derivatives with modifications on a phthalimide ring (NE-001 to NE-010; Fig. 1a). An AlphaScreen-based interaction assay using a wheat cell-free system (Fig. 1b) established in our previous studies[10,35] showed that thalidomide, lenalidomide, pomalidomide, and 5-hydroxythalidomide showed a stronger or equal ability to interact with SALL4/PLZF than with IKZF1 (Fig. 1c). Alternatively, 6-fluoro lenalidomide (NE-005) interacted more strongly with IKZF1 than with SALL4 and PLZF (Fig. 1c). Furthermore, immunoblot analysis showed that NE-005 induced the protein degradation of

exogenous Myc-IKZF1 but not AGIA-SALL4 (Fig. 1d). By contrast, 5-fluoro lenalidomide (NE-008) and 7-fluoro lenalidomide (NE-006) barely induced interactions between CRBN neosubstrates and could not degrade Myc-IKZF1 and AGIA-SALL4 (Fig. 1c, d). 6-fluoro pomalidomide (NE-003) strongly induced protein degradation of both Myc-IKZF1 and SALL4 (Fig. 1d). Furthermore, NE-005 induced selective and strong degradation of endogenous IKZF1, IKZF3, and CK1α, but not SALL4 and PLZF, in cultured cells (Fig. 1e, f). Notably, NE-005 induced drastic degradation of CK1α, which is involved in the anti-5q MDS activity of lenalidomide[7], at 100-fold lower concentrations than those of lenalidomide (Fig. 1e, f). Therefore, we synthesised 6-position-modified lenalidomide and pomalidomide (NE-011–NE-014) using halogen atoms (Fig. 1g). AlphaScreen-based interaction assays showed that 6-chloro lenalidomide (NE-013) selectively interacted with IKZF1, whereas 6-bromo lenalidomide (NE-014) scarcely interacted with neosubstrates (Fig. 1h). Conversely, the 6-position-modification on pomalidomide did not increase selectivity for IKZF1 and reduced binding ability towards IKZF1, SALL4, and PLZF (Fig. 1h). These results suggest that 6-position-modification of lenalidomide is a viable approach for enhancing IKZF1 selectivity over SALL4. Previous studies showed that both the binding and protein degradation abilities toward SALL4 of lenalidomide are lower than those of thalidomide and pomalidomide[8,9]. Lenalidomide is the first-line treatment for MM and 5q MDS and is the most widely used IMiD[1–3]. Therefore, modifying lenalidomide to increase selectivity for anti-haematological cancer activity is reasonable.

We then characterised the 6-position-modified lenalidomide using biochemical and cell-based experiments. The biochemical interaction assay revealed that 6-fluoro lenalidomide (NE-005/F-Le) interacted with SALL4 at the same level as lenalidomide (Le) but interacted more strongly with IKZF1 than Le (Fig. 2a, b). However, 6-chloro lenalidomide (NE-013/Cl-Le) did not interact with SALL4, but its affinity for IKZF1 was lower than that for Le (Fig. 2b). These differences in the binding ability for neosubstrates to CRBN were also validated via an in vitro pull-down assay using recombinant proteins (Supplementary Fig. 1a). In addition, the in vitro ubiquitination assay showed that the polyubiquitination level of SALL4 by F-Le was weaker than that of Le, while the polyubiquitination level of IKZF1 was the same as that of Le (Fig. 2c). Consistent with results of the in vitro ubiquitination assay, the cellular polyubiquitination of SALL4 by F-Le was weak (Fig. 2d). However, cellular polyubiquitination of IKZF1 was the strongest in HEK293T cells (Fig. 2d). We then investigated the degradation of neosubstrates by lenalidomide derivatives using cell lines expressing these neosubstrates. Immunoblot analyses confirmed that F-Le strongly induced the protein degradation of IKZF1, IKZF3, and CK1α (Fig. 2e, f), but the induction of SALL4 degradation by F-Le was weaker than that by Le (Fig. 2f, g, Supplementary Fig. 1b). Cl-Le was more selective, but its degradation ability for IKZF1, IKZF3, and CK1α was lower than that of Le (Fig. 2e–g, Supplementary Fig. 1b). 6-bromo lenalidomide (Br-Le) barely induced the protein degradation of IKZF1, IKZF3, and CK1α (Fig. 2e–g, Supplementary Fig. 1b). Minor changes in substituent size at the 6-position of lenalidomide significantly alter neosubstrate selectivity, and 6-fluoro lenalidomide and 6-chloro lenalidomide may be more selective thalidomide derivatives for neosubstrates involved in anti-haematological cancer activity.

### 6-fluoro and 6-chloro lenalidomides show anti-proliferative effects on MM and 5q MDS
The 6-position-modified lenalidomides induced selective protein degradation of neosubstrates involved in anti-MM and anti-5q MDS activities (Figs. 1, 2). Therefore, we investigated whether lenalidomide derivatives exerted anti-proliferative effects on MM and 5q MDS cell lines. Cell-Titer Glo assays showed that F-Le strongly reduced cell growth in the Le-sensitive MM cell lines MM1.S, H929, and U266, but not in the Le-insensitive cell line RPMI8226[5,6,36] (Supplementary

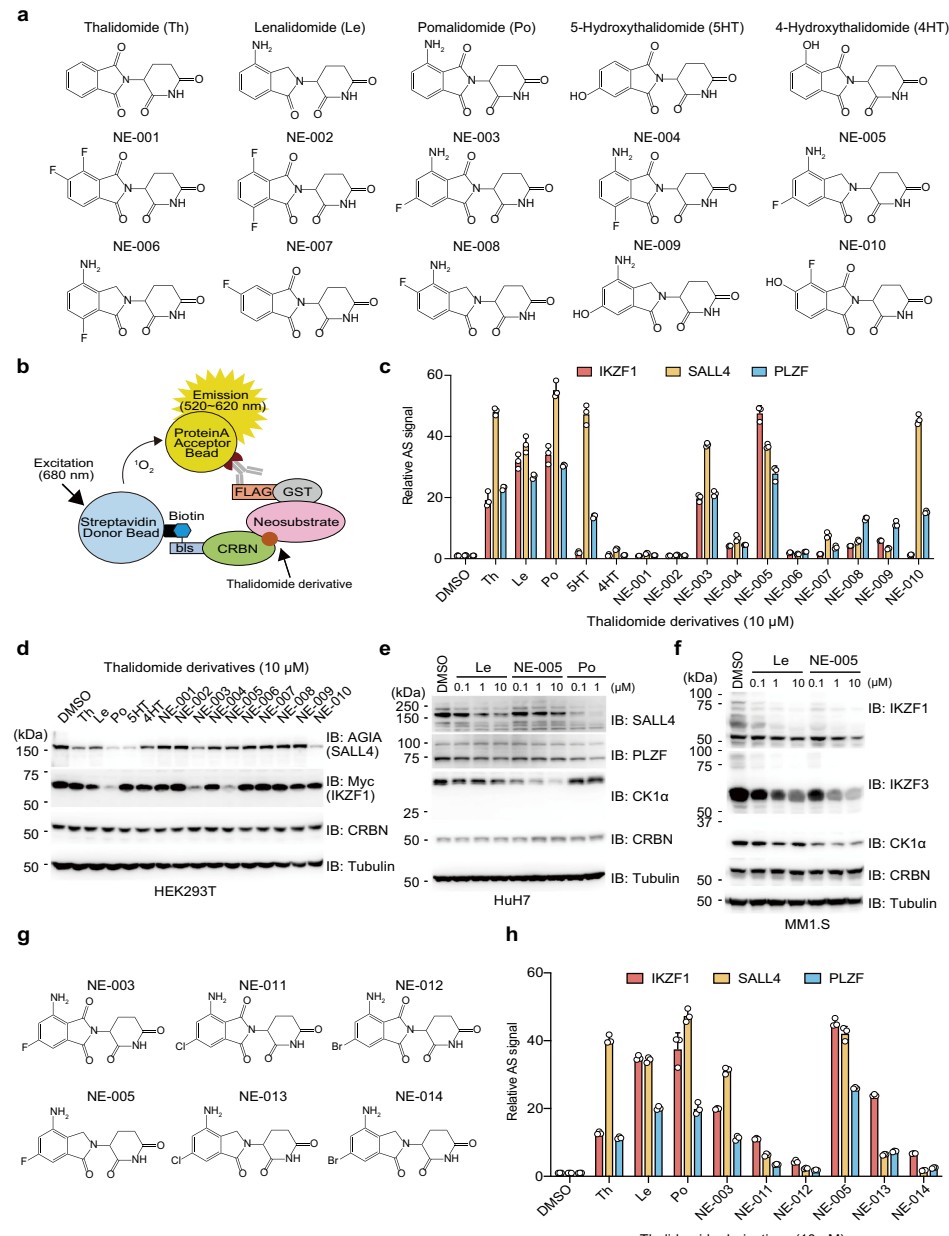

**Fig. 1 | Identification of critical positions on thalidomide derivatives for enhancing IKZF1 selectivity over SALL4. a** Chemical structures of thalidomide (Th), lenalidomide (Le), pomalidomide (Po), 5-hydroxythalidomide (5HT), 4-hydroxythalidomide (4HT), and 10 thalidomide derivatives (NE-001–NE-010). **b** Schematic diagram of the AlphaScreen (AS)-based biochemical interaction assay using recombinant proteins for detecting thalidomide-derivative-dependent complex formation between CRBN and neosubstrates. **c** Thalidomide-derivative-dependent biochemical interaction assay. The CRBN–IKZF1, CRBN–SALL4, and CRBN–PLZF complex formation was analysed using AS technology. The relative AS signals are expressed as the luminescence signal relative to the luminescence signal of dimethylsulfoxide (DMSO), which is considered 1. Error bars denote standard deviations (independent experiments, $n = 3$). **d** Immunoblot analysis of thalidomide-derivative-dependent exogenous neosubstrate degradation. HEK293T cells were transfected with pcDNA3.1-AGIA-SALL4 and pcDNA3.1-Myc-

IKZF1 and treated with DMSO, 10-μM Th, or 10-μM thalidomide derivatives for 16 h. The experiment was independently repeated thrice, with similar results. **e–f** Immunoblot analysis of dose-dependent endogenous neosubstrate degradation in (**e**) HuH7 cells or (**f**) MM1.S cells. Each cell line was treated with DMSO, Po, Le, or NE-005 for 24 h. The experiment was independently repeated thrice, with similar results. **g** Chemical structures of 6-halogenated pomalidomides (NE-003, NE-011, and NE-012) and 6-halogenated lenalidomides (NE-005, NE-013, and NE-014). **h** Thalidomide-derivative-dependent biochemical interaction assay. The CRBN–IKZF1, CRBN–SALL4, and CRBN–PLZF complex formation was analysed using AS technology. The relative AS signals are expressed as the luminescence signal relative to the luminescence signal of DMSO, which is considered 1. Error bars denote standard deviations (independent experiments, $n = 3$). Source data are provided as a Source data file.

Fig. 2a). In the Le-sensitive 5q MDS-L cell line[37], F-Le showed a stronger anti-proliferative effect than Le, and Cl-Le at the same level as Le (Supplementary Fig. 2b). It has been reported that the down-regulation of IRF4 and MYC via IKZF1 degradation is a key mechanism underlying the anti-proliferative effect of Le and pomalidomide (Po) in MM cells[5,6,36]. Immunoblot analysis showed that F-Le and Cl-Le

treatments slightly reduced the protein expression level of IRF4 and down-regulated MYC in MM1.S and H929 cell lines (Fig. 3a). Regarding the mechanism of action of lenalidomide in 5q MDS cells, the protein degradation of both IKZF1 and CK1α was required for the anti-proliferative effect of lenalidomide[7,37]. First, IKZF1 degradation by lenalidomide induced up-regulation of RUNX1, followed by induction

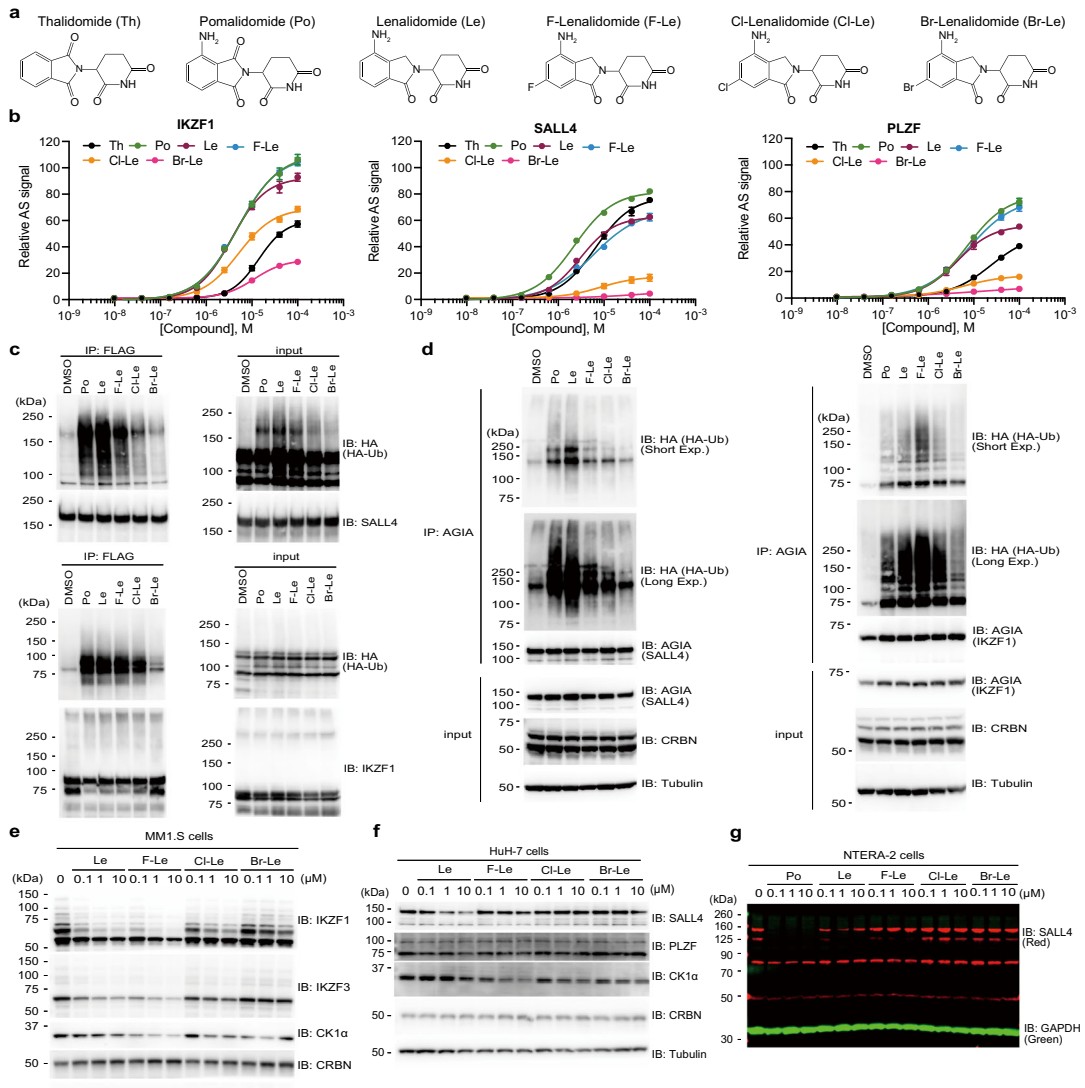

**Fig. 2 | Biochemical and cell-based analyses of 6-position-modified lenalido-mides. a** Chemical structures of thalidomide (Th), pomalidomide (Po), lenalido-mide (Le), 6-fluoro lenalidomide (F-Le), 6-chloro lenalidomide (Cl-Le) and 6-bromo lenalidomide (Br-Le). **b** Dose-dependent interaction assay using AlphaScreen (AS) technology. The CRBN–IKZF1, CRBN–SALL4, and CRBN–PLZF complex formation was analysed. The relative AS signals are expressed as the luminescence signal relative to the luminescence signal of dimethylsulfoxide (DMSO), which is con-sidered 1. Error bars denote standard deviations (independent experiments, *n* = 3). **c** In vitro ubiquitination assay of SALL4 and IKZF1 by CRL4$^{CRBN}$. Purified FLAG-GST-IKZF1 or -SALL4 were mixed with recombinant CRL4$^{CRBN}$, E1, E2, and HA-Ub, and ubiquitination reactions were performed in the presence of DMSO, 20-μM poma-lidomide (Po), 20-μM Le, 20-μM F-Le, 20-μM Cl-Le, or 20-μM Br-Le. Ubiquitinated

SALL4 and IKZF1 were immunoprecipitated using an anti-FLAG antibody. The experiment was repeated twice independently, with similar results. **d** In the cell ubiquitination assay of SALL4 and IKZF1 by CRL4$^{CRBN}$. HEK293T cells were trans-fected with pcDNA3-HA-ubiquitin and pcDNA3.1-FLAG-CRBN and pcDNA3.1-AGIA-SALL4 or -IKZF1 and treated with DMSO, 1-μM Po, 10-μM Le, 10-μM F-Le, 10-μM Cl-Le, or 10-μM Br-Le in the presence of 10-μM MG132 for 8 h. Ubiquitinated SALL4 and IKZF1 were immunoprecipitated using an anti-AGIA antibody. The experiment was repeated twice independently, with similar results. **e–g** Immunoblot analysis of dose-dependent neosubstrate degradation in (**e**) MM1.S, (**f**) HuH7, or (**g**) NTERA-2 cells. Each cell line was treated with DMSO, Po, Le, F-Le, Cl-Le, or Br-Le for 24 h. The experiment was independently repeated thrice, with similar results. Source data are provided as a Source data file.

of differentiation into megakaryocytes[37]. Subsequently, lenalidomide induced the protein degradation of CK1α, which has a low expression level in 5q MDS cells, and induced apoptosis[37]. Po, Le, F-Le, and Cl-Le induced the protein degradation of IKZF1 and up-regulation of RUNX1 (Fig. 3b) and increased the mRNA expression levels of *SELP* and *ITGB3*, which are induced by lenalidomide for differentiation into megakaryocytes[37] (Fig. 3c). These results suggest that lenalidomide derivatives have the same mechanism of action as lenalidomide and pomalidomide.

We then used dose-dependent analyses to compare the anti-proliferative effects of lenalidomide, pomalidomide, and lenalido-mide derivatives. F-Le and Cl-Le showed dose-dependent anti-

proliferative effects on MM and 5q MDS cell lines (Supplementary Fig. 2c and d) but Br-Le did not (Supplementary Fig. 2c, d). Next, we quantitatively analysed the anti-proliferative effect of F-Le and Cl-Le. F-Le exerted a stronger anti-proliferative effect on both MM and 5q MDS cell lines than did Le (Fig. 3d–f). Cl-Le also showed a weaker anti-proliferative effect on the cell lines than did Le (Fig. 3d–f). Importantly, F-Le showed higher efficiency in MDS-L cells than did lenalidomide (Fig. 3f), consistent with the degradation ability of F-Le for CK1α (Fig. 3a, b). In previous studies, CC-122 and CC-90009, which are thalidomide derivatives, also showed an anti-proliferative effect on diffuse large B-cell lymphoma (DLBCL)[14] and acute myeloid leukaemia (AML)[13], respectively. Therefore, we then investigated

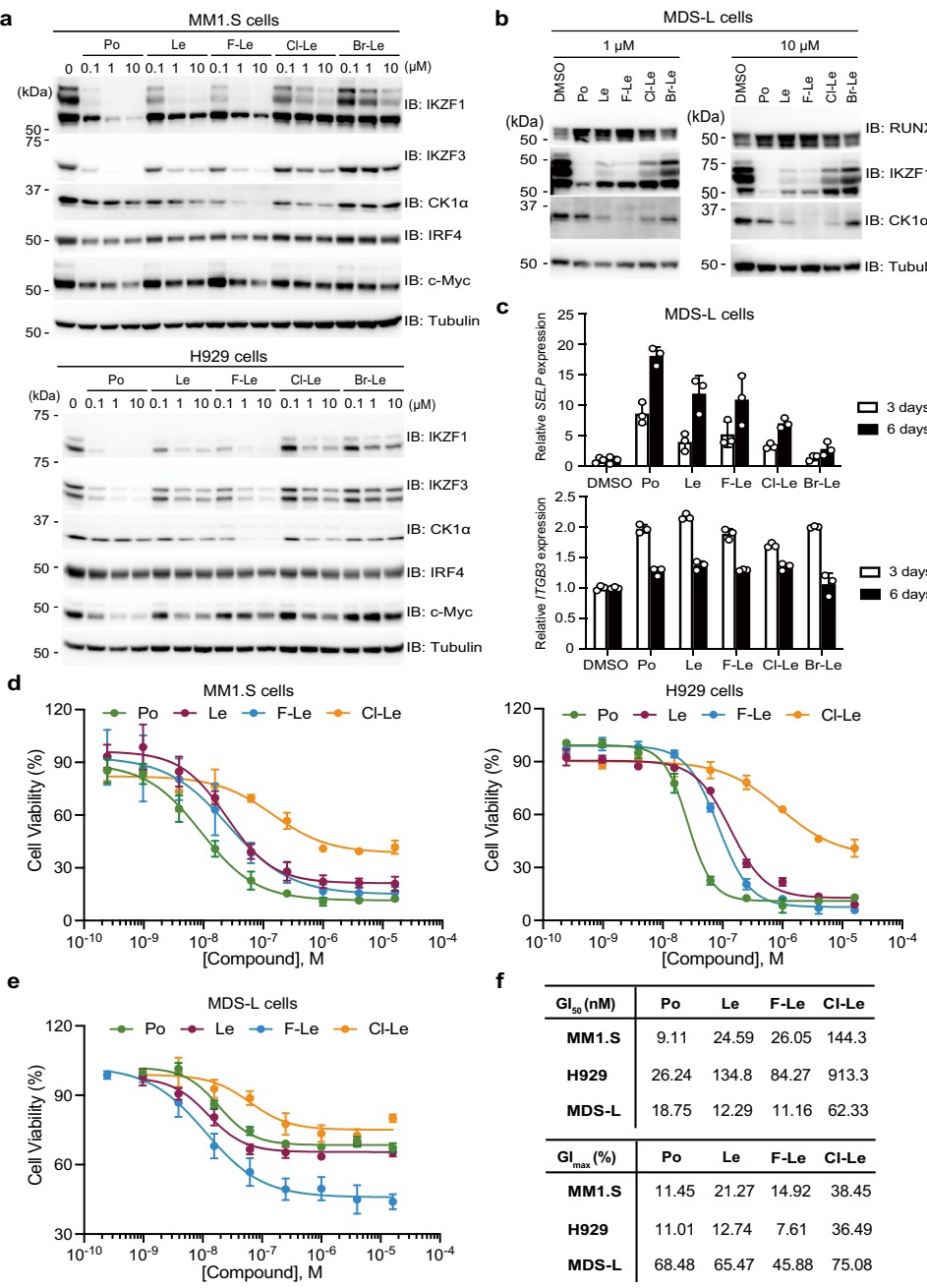

**Fig. 3 | Anti-proliferative effect of 6-position-modified lenalidomide on multiple myeloma and 5q myelodysplastic syndromes. a** Immunoblot analysis of IRF4 and c-Myc. MM1.S or H929 cells were treated with dimethylsulfoxide (DMSO), Po, Le, or lenalidomide derivatives for 72 h. The experiment was independently repeated thrice, with similar results. **b** Immunoblot analysis of RUNX1. Myelodysplastic syndrome (MDS)-L cells were treated with DMSO, Po, Le, or lenalidomide derivatives for 24 h. The experiment was repeated twice independently, with similar results. **c** Expression of *SELP* and *ITGB3*, which are upregulated in 5q MDS cells treated with lenalidomide. MDS-L cells were treated with DMSO, 10-μM Po, 10-μM Le, or 10-μM lenalidomide derivatives for 3 or 6 days, and mRNA expression was measured using quantitative RT-PCR. Relative mRNA expression was determined using the expression level following DMSO treatment. Error bars denote standard deviation (biological replicates; $n = 3$). **d** Dose–response curve of the anti-proliferative effect of 6-position-modified lenalidomides on MM cell lines. MM1.S

and H929 cells were treated with DMSO, Po, Le, F-Le, or Cl-Le for 10 days, and cell viability was analysed using the CellTiter-Glo assay kit. Cell viability was expressed as the luminescence signal relative to the luminescence signal of DMSO, which was considered 100. Error bars denote standard deviation (biological replicates; $n = 3$). **e** Dose–response curve of anti-proliferative effect by 6-position-modified lenalidomides on 5q MDS cell line. MDS-L cells were treated with DMSO, Po, Le, F-Le, or Cl-Le for 12 days, and cell viability was analysed using the CellTiter-Glo assay kit. Cell viability was expressed as the luminescence signal relative to the luminescence signal of DMSO, which was considered 100. Error bars denote standard deviation (biological replicates; $n = 4$). **f** The half-maximal growth inhibition ($GI_{50}$) and the maximal growth inhibition ($GI_{max}$) values were calculated using the dose–response curve in (**d**, **e**). N/A means not applicable. Source data are provided as a Source data file.

whether F-Le and Cl-Le showed an anti-proliferative effect on haematological cancer cell lines other than MM and 5q MDS cells. CellTiter Glo revealed that F-Le and Cl-Le did not have significant cytotoxicity on TK (B-cell non-Hodgkin lymphoma), BJAB (Burkitt lymphoma), Karpas-1106P (Primary mediastinal DLBCL), SU-DHL-4 (germinal centre B cell-like DLBCL), Jurkat (childhood T acute lymphoblastic leukaemia) and THP-1 (Childhood acute monocytic leukaemia) cells (Supplementary Fig. 3a–f). These results suggest that

6-fluoro and 6-chloro lenalidomides are selective and highly effective lenalidomide derivatives for treating MM and 5q MDS.

### Evaluation of structure–activity relationship of 6-position-modified lenalidomide

The results shown in Figs. 1–3 suggested that the ability of 6-position-modified lenalidomides for neosubstrate degradation decreased, as 6-position was modified with bulky substituent. To investigate the structure–activity relationship (SAR) of 6-modified lenalidomides, we synthesised 6-trifluoromethyl lenalidomide (NE-015/F$_3$C-Le) and 6-trifluoromethoxy lenalidomide (NE-016/F$_3$CO-Le) (Fig. 4a). An AlphaScreen-based interaction assay showed that F$_3$C-Le and F$_3$CO-Le could not interact with IKZF1, SALL4, and PLZF (Fig. 4b). The cellular binding abilities toward the neosubstrates were validated using streptavidin pull-down assay (STA-PDA) based on the proximity-dependent biotin identification (BioID) method, which was established in our previous studies[30,38], and it was shown that F$_3$C-Le and F$_3$CO-Le scarcely biotinylated SALL4, PLZF, IKZF1, and IKZF3 (Fig. 4c). Furthermore, immunoblot analyses showed that F$_3$C-Le and F$_3$CO-Le could not degrade IKZF1, IKZF3, CK1α, SALL4, and PLZF in cultured cells (Fig. 4d–f). Then, to quantitatively analyse the ability of 6-position-modified lenalidomides for neosubstrate degradation, we

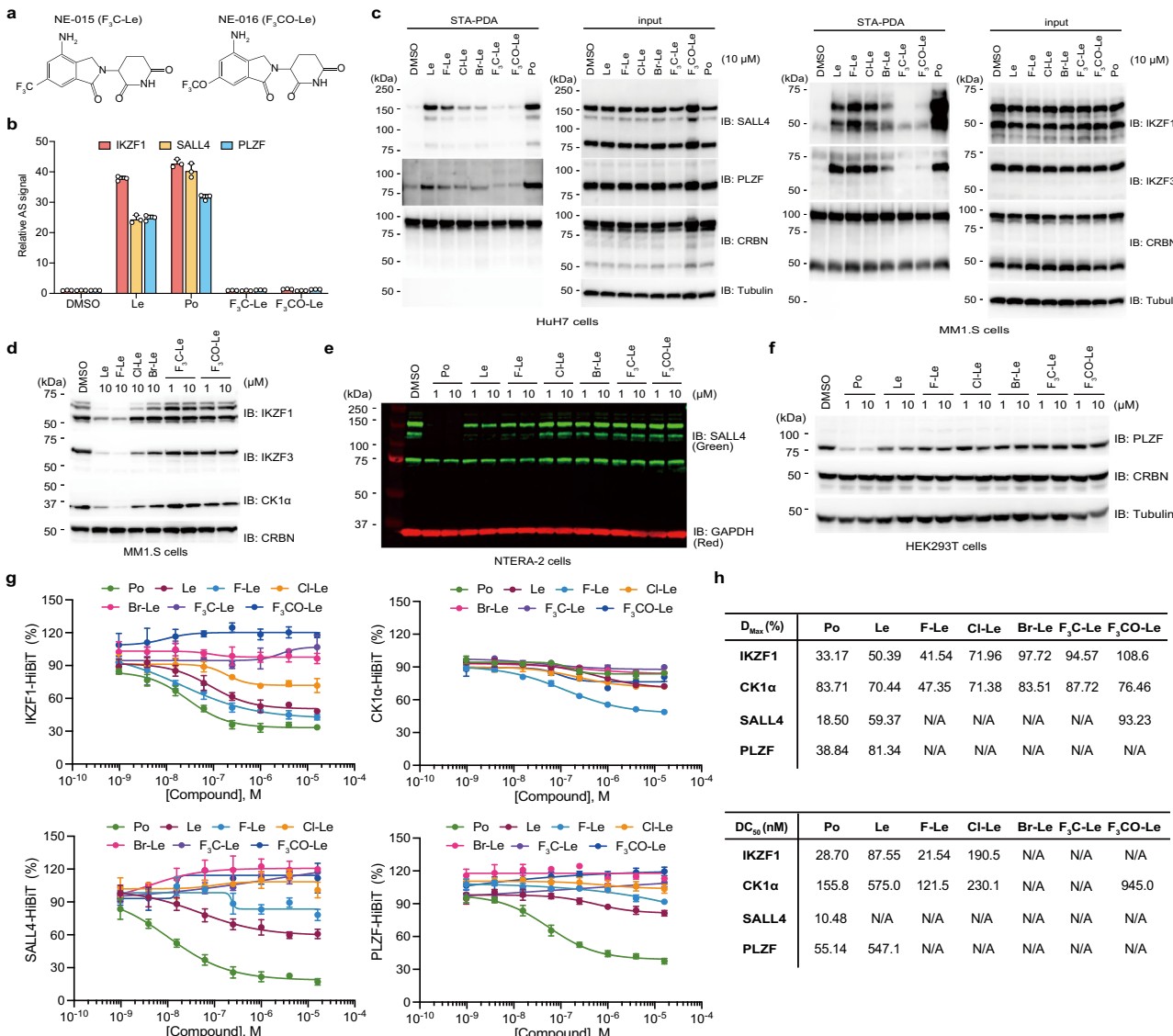

**Fig. 4 | Neosubstrate selectivity for IKZF1, CK1α, SALL4, and PLZF of 6-position-modified lenalidomides. a** Chemical structures of NE-015 (F$_3$C-Le) and NE-016 (F$_3$CO-P). **b** Thalidomide-derivative-dependent biochemical interaction assay. The CRBN–IKZF1, CRBN–SALL4, and CRBN–PLZF complex formation was analysed using AlphaScreen (AS) technology. The relative AS signals are expressed as the luminescence signal relative to the luminescence signal of dimethylsulfoxide (DMSO), which is considered 1. Error bars denote standard deviations (independent experiments, $n = 3$). **c** In-cell proximity-dependent biotinylation of neosubstrates by AirID-CRBN. HuH7 and MM1.S cells stably expressing AGIA-AirID-CRBN were treated with DMSO, Po, Le, or lenalidomide derivatives in the presence of 10-μM biotin and 5-μM MG132 for 6 h. The experiment was repeated twice independently, with similar results. **d–f** Immunoblot analysis of degradations of five neosubstrates in (**d**) MM1.S cells, (**e**) NTERA-2 cells, or (**f**) HEK293T cells. Each cell line was treated with DMSO, Po, Le, or lenalidomide derivatives for 24 h. The experiment was independently repeated thrice, with similar results. **g** Dose–response curves of degradation of four neosubstrates by 6-position-modified lenalidomides using HiBiT system. HEK293T cells stably expressing IKZF1, CK1α, SALL4, or PLZF with C-terminal HiBiT-tag were treated with DMSO, Po, Le, or lenalidomide derivatives for 16 h. The protein expression level was expressed as the luminescence signal relative to the luminescence signal of DMSO, which was considered 100. Error bars denote standard deviation (biological replicates; $n = 3$). **h**, The half-maximal degradation concentration (DC$_{50}$) and the maximal degradation ($D_{max}$) values were calculated using dose–response curves in (**g**). N/A means not applicable. Source data are provided as a Source data file.

generated HEK293T cells stably expressing neosubstrate-HiBit using lentivirus. The luminescence signal of IKZF1-HiBit was markedly reduced by Po, Le, and F-Le in a dose-dependent manner (Fig. 4g) and slightly reduced by Cl-Le (Fig. 4g). In the case of CK1α-HiBit, F-Le induced a more robust protein degradation of CK1α-HiBit than did Le (Fig. 4g). Furthermore, the luminescence signals of SALL4-HiBit and PLZF-HiBit were markedly reduced by Po and slightly reduced by Le (Fig. 4g). Importantly, the degradation abilities of F-Le and Cl-Le were lower than that of Le, and reductions of the luminescence signals of the four neosubstrates-HiBit by Br-Le, $F_3$C-Le, and $F_3$CO-Le were scarcely observed (Fig. 4g). In addition, the degradation abilities of 6-position-modified lenalidomides for the four neosubstrates were quantified using $DC_{50}$ and $D_{max}$ (Fig. 4h), supporting the SAR shown in Figs. 1–3 of the 6-position-modified lenalidomides.

Next, to understand the molecular basis of the change in neosubstrate selectivity induced by a substituent at the 6-position of the phthalimide ring, we analysed the docking poses of 6-position-modified lenalidomides at the IMiDs-binding sites of IKZF1–CRBN, SALL4–CRBN and CK1α–CRBN complexes[33,35,39]. The reported crystal structures of these complexes show that the three major IMiDs, Th, Le and Po, all adopt the same binding modes and that the 6-CH group of the phthalimide ring is oriented toward the β-hairpin structure of each neosubstrate due to the interaction of the 4-amino group of Le and Po with the E377 residue of CRBN (Supplementary Fig. 4a). In our docking simulation using AutoDock Vina[40] (The Scripts Research Institute, version 1.1.2), the best docking pose of each 6-position-modified lenalidomide with the lowest affinity score (third docking pose of $F_3$C-Le to the SALL4-CRBN complex) was predicted to be almost the same binding mode as Le (Supplementary Fig. 4b and Supplementary Table 1). All the 6-position substituents were located in the space bound by the H353 residue of CRBN and some residues of the neosubstrates (Q146 and N148 for IKZF1, V411 and S413 for SALL4, and K18, I35, and I37 for CK1α). However, the orientation change of these residues, especially the H353 residue of CRBN, was required to allow larger substituents such as $F_3$C and $F_3$CO groups in the space, which may have a disruptive effect on the contact between CRBN and the neosubstrates. The comparative docking poses also raise the possibility that the different types and combinations of residues located around the 6-position substituent among the three neosubstrates contribute to the neosubstrate selectivity (Supplementary Fig. 4b). For example, there is a polar residue distal to the H353 residue of CRBN across the 6-fluoro or 6-chloro group on the phthalimide group in IKZF1 (Q146) and CK1α (K18), whereas no such residue was found in SALL4. Taken together, our experimental results and docking analyses propose that modification of 6-position on phthalimide ring with small substituent increases the selectivity towards IKZF1, IKZF3 and CK1α, and modification with bulky substituent eliminates degradation ability.

## Evaluation of selectivity of 6-position-modified lenalidomides towards global neosubstrates

Previous studies showed that many neosubstrates, such as ZFP91[41] and RAB28[9], are degraded by thalidomide derivatives. In addition, many thalidomide derivatives and neosubstrate pairs, such as CC-885–GSPT1[8] and CC-122–ZMYM2[42], were reported. Therefore, we investigated the protein degradation of various neosubstrates by 6-position-modified lenalidomides. Consistent with the results of previous studies, ZMYM2 was degraded by Po[30] and CC-122[42] (Fig. 5a). In addition, Po, Le, and F-Le induced protein degradation of ZFP91 but Cl-Le, Br-Le, $F_3$C-Le, and $F_3$CO-Le did not (Fig. 5a), and all 6-position-modified lenalidomides did not degrade GSPT1, which was degraded by CC-885 (Fig. 5b and Supplementary Fig. 5a). Furthermore, the observed neosubstrate selectivity was also validated in HuH7 cells (Fig. 5b). In addition to novel thalidomide derivatives, several CRBN modulators having glutarimide rings were developed, and ALV2 induced protein degradation of IKZF2[43]. More recently, it was reported

that NVP-DKY709 also is an IKZF2 degrader[44]. Therefore, we examined whether 6-position-modified lenalidomide induced protein degradation of IKZF2. Immunoblot analysis showed that all 6-position-modified lenalidomides could not degrade IKZF2 (Fig. 5c). Alternatively, F-Le, Cl-Le, Br-Le, and $F_3$C-Le inued protein degradation of RAB28, which was reported as a Po/Le neosubstrate[9] (Fig. 5c). In addition, 6-position-modified lenalidomides slightly induced protein degradation of RNF166, which is a pan-IMiD neosubstrate[9] (Supplementary Fig. 5b).

Thalidomide-induced teratogenicity occurs in thalidomide-sensitive species, including non-human primates[45], rabbits[8,46] and chickens[4]. However, rodents including mice and rats are known as non-sensitive species to thalidomide[8,47]. This species-specificity was caused by differences in amino acid sequences of CRBN among species[7,9,10]. SALL4[8] and PLZF[10] are neosubstrates involved in thalidomide-induced teratogenicity, as shown using rabbit and chicken models, respectively. p63 is a neosubstrate involved in thalidomide teratogenicity in zebrafish[48], but the use of the zebrafish model to analyse limb teratogenicity induced by thalidomide derivatives is controversial[49]. Importantly, the 6-position-modified lenalidomides did not induce protein degradation of p63 (TP63) (Fig. 5d). Immunoblot analyses showed that human CRBN (HsCRBN) can induce SALL4 degradation, but zebrafish Crbn (DrCrbn) cannot (Supplementary Fig. 5c, d), indicating that a wild type zebrafish model cannot be used to precisely evaluate thalidomide-induced teratogenicity. Furthermore, chicken Crbn also cannot induce protein degradation of chicken Sall4[10]. Therefore, to precisely analyse thalidomide-induced teratogenicity in vivo, it is necessary to establish a mammalian model. A previous study reported that the level of MEIS2, which is an endogenous CRBN substrate and involved in limb development, was increased by IMiD treatment[31]. Immunoblot analysis showed that all 6-position-modified lenalidomides slightly increased the expression level of MEIS2 at the same level as that of Le and Po treatment (Fig. 5e).

Finally, to investigate more global protein degradations of neosubstrates, we performed 18-plex tandem mass tag (TMT) labelling and mass spectrometry (MS) analysis (Supplementary Fig. 6a–h and Supplementary Data 1). As shown in Supplementary Fig. 6, thalidomide (Th) did not induce drastic protein degradation of neosubstrates reported in previous studies, but several proteins were reduced (described by grey characters in Supplementary Fig. 6a). By contrast, MM1.S cell lysates treated with Po reduced many neosubstrates, such as DTWD1, IKZF1, IKZF3 and ZFP91 (Supplementary Fig. 6b). Le, F-Le and Cl-Le induced protein degradation of CK1α and RAB28 in addition to IKZF1 and IKZF3 (Supplementary Fig. 6c–e). Br-Le and $F_3$C-Le degraded RAB28 but did not CK1α, IKZF1 and IKZF3 (Supplementary Fig. 6f, g). $F_3$CO-Le did not induce protein degradation of detected neosubstrates in MM1.S cells (Supplementary Fig. 6h). The protein levels of these neosubstrates were showed by heatmap (Fig. 6f), supporting our hypothesis that 6-position modification with bulky molecule, such as $F_3$C and $F_3$CO, remarkably decreased neosubstrate degradation ability of 6-position-modified lenalidomide.

## Degradation of target proteins and neosubstrates by PROTACs based on 6-modified lenalidomides

IMiD-based PROTACs using various target binders have been developed[19,20], and PROTACs induced protein degradation of IMiD neosubstrates[20,29]. IMiD-based PROTACs (Supplementary Fig. 7a) induced protein degradation of neosubstrates with different neosubstrate selectivities (Supplementary Fig. 7b, c). Therefore, we investigated whether 6-position-modified lenalidomides can be used as CRBN binders for PROTACs. To examine the binding ability of 6-position-modified lenalidomides to CRBN, we generated thalidomide-immobilised magnetic beads using a thalidomide derivative (Supplementary Fig. 8a). Recombinant FLAG-GST-CRBN was pulled down using thalidomide-immobilised beads, followed by competitively

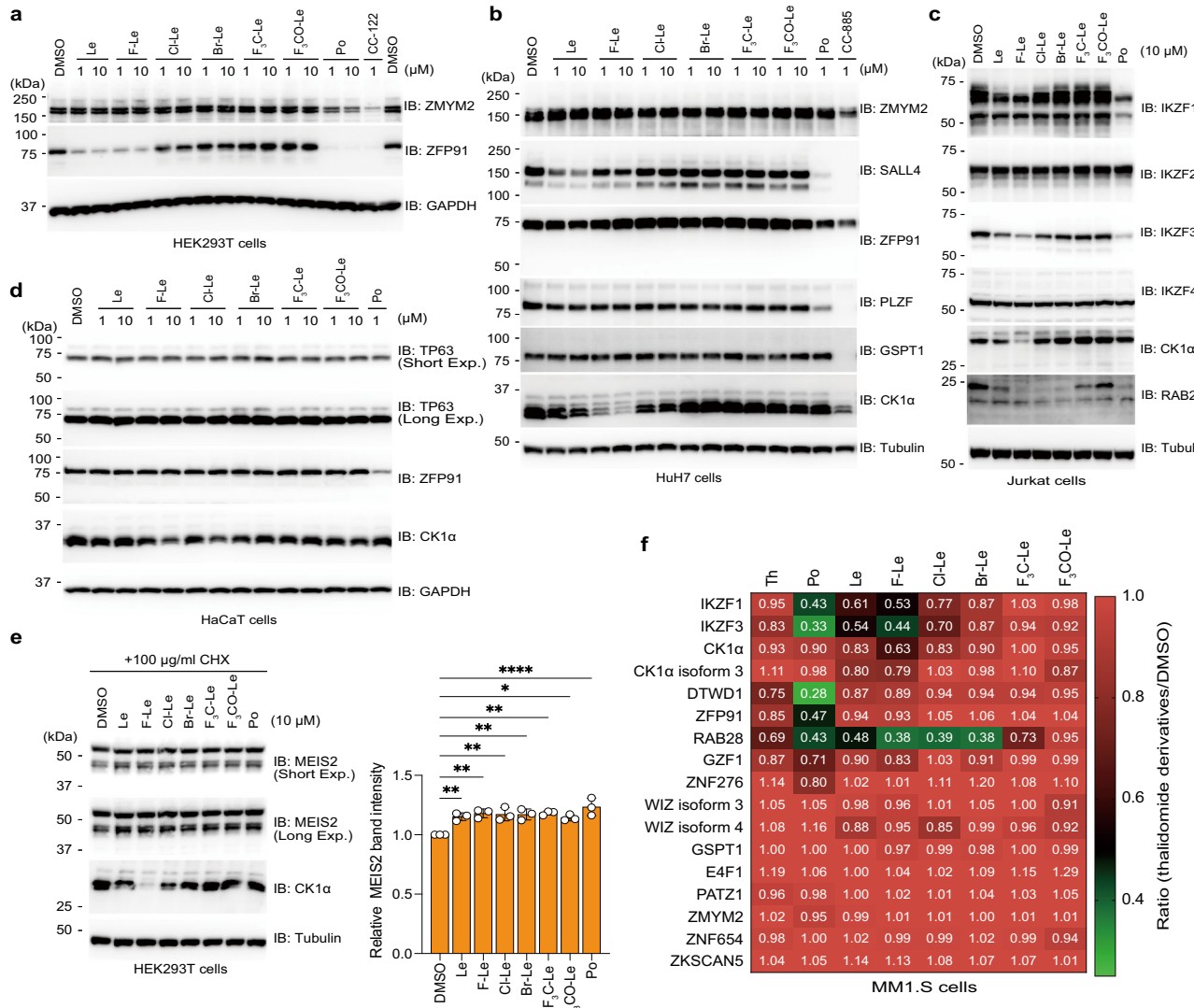

**Fig. 5 | Evaluation of global neosubstrate selectivity using 6-position-modified lenalidomides. a** Neosubstrate selectivity of 6-position-modified lenalidomides in HEK293T cells. HEK293T cells were treated with dimethylsulfoxide (DMSO), Po, Le, lenalidomide derivatives, or CC-122 for 48 h. The experiment was independently repeated thrice, with similar results. **b**–**d** Neosubstrate selectivity of 6-position-modified lenalidomides in HuH7, Jurkat, and HaCaT cells. (**b**) HuH7, (**c**) Jurkat, or (**d**) HaCaT cells were treated with DMSO, Po, Le, lenalidomide derivatives, CC-122, or CC-885 for 24 h. The experiment was independently repeated twice, with similar results. **e** Effect of 6-position-modified lenalidomides on MEIS2 protein expression level. HEK293T cells were treated with DMSO, Po, Le, or lenalidomide derivatives in

the presence of cycloheximide (CHX) for 4 h. All relative MEIS2 band intensities are expressed as band intensity relative to that of DMSO. Error bars denote the standard deviation (biological replicates; n = 3), and P-values were calculated using one-way ANOVA with Tukey's post-hoc test (*P < 0.05, **P < 0.01, and ****P < 0.0001). **f** Heatmap showing the ratio of global neosubstrates in whole-proteome quantification. MM1.S cells were treated with DMSO, 10-μM Th, 10-μM Po, 10-μM Le, or 10-μM lenalidomide derivatives for 5 h; quantitative proteomics analysis was performed using a tandem mass tag (biological replicates, n = 2). Source data are provided as a Source data file.

eluting with 6-position-modified lenalidomides. Immunoblot analysis revealed that the 6-position-modified lenalidomides interacted with CRBN (Supplementary Fig. 8b). Then, we performed competition experiments in an AlphaScreen-based interaction assay using biotinylated thalidomide (Supplementary Fig. 8c). All 6-position-modified lenalidomides decreased luminescence signals in a dose-dependent manner (Supplementary Fig. 8d) and showed similar binding ability as that of Le and Po (Supplementary Fig. 8e). To investigate the binding ability of 6-position-modified lenalidomides quantitatively, we performed isothermal titration calorimetry (ITC) analysis, which was established in a previous study[35]. The $K_D$ values showed that the affinities of F-Le, Cl-Le, Br-Le, and F_3CO-Le were 2-3 times lower than that of Le (Supplementary Fig. 8f). However, we could not evaluate the binding ability of F_3C-Le because the interaction between CRBN and F_3C-Le was not detected under ITC assay conditions (Supplementary

Fig. 8f). Given that the affinity of F_3CO-Le ($K_D = 2.83 \pm 0.17$ μM) was the highest among those of the 6-position-modified lenalidomides and higher than that of CRBN−(S)-thalidomide ($K_D = 4.00 \pm 0.36$ μM[35]), 6-position modification with a bulky molecule did not significantly affect the binding ability to CRBN. These results indicate that 6-position-modified lenalidomides can interact with CRBN and can be used as CRBN binders for PROTACs.

Next, we purchased or synthesised PROTACs for BET proteins using a BET inhibitor (OTX-015) and pomalidomide (ARV-825/Po-P), lenalidomide (Le-P) or 6-position-modified lenalidomides (F-P, Cl-P and F_3C-P) (Fig. 6a). Consistent with Supplementary Fig. 8d, the PROTACs based 6-position-modified lenalidomides interacted with CRBN (Fig. 6b). The IC_{50} values revealed that F-P and Cl-P interacted with CRBN in the same dose range as Le-P, and the binding ability of F_3C-P was about two times lower than that of Le-P (Fig. 6b). An AlphaScreen-

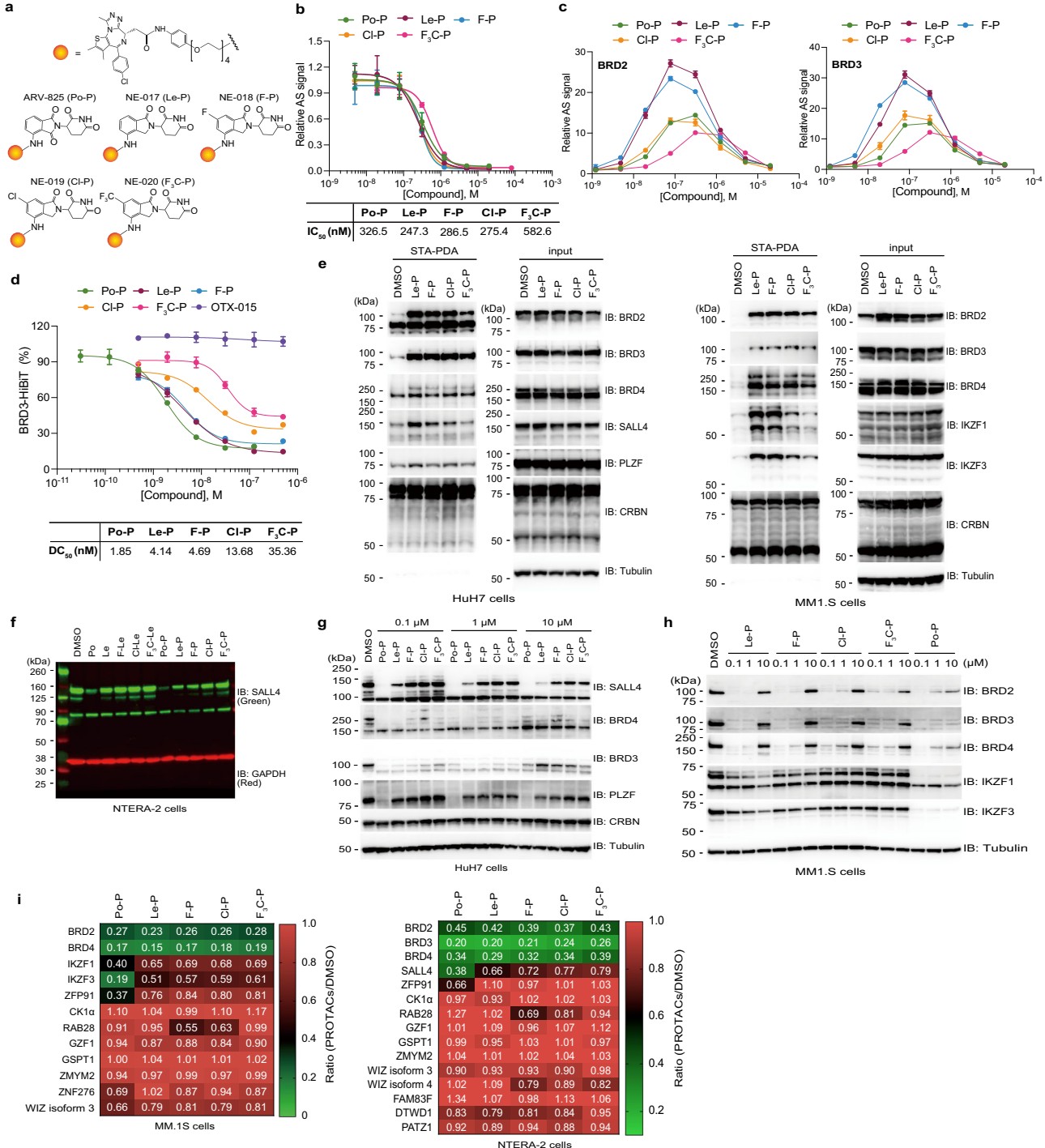

**Fig. 6 | Neosubstrate selectivity of PROTACs using 6-position-modified lenalidomides. a** Chemical structures of ARV-825 (Po-P), NE-017 (Le-P), NE-018 (F-P), NE-019 (Cl-P), and NE-020 (F₃C-P). **b** Binding ability of proteolysis-targeting chimeras (PROTACs) for CRBN was analysed using AlphaScreen (AS) technology. The half-maximal inhibition concentration (IC50) values were calculated using the dose–response curves in (**b**). **c**, The ternary complex formation was analysed using AS technology. **d** Dose–esponse curves of degradation of four neosubstrates using HiBiT system. HEK293T cells stably expressing BRD3-HiBiT were treated with dimethylsulfoxide (DMSO) or PRTOACs for 6 h. Protein expression was expressed as the luminescence signal relative to the luminescence signal of DMSO, which was considered 100. Error bars denote standard deviation (biological replicates; n = 3). The DC50 values were calculated using the dose–response curves in (**d**). **e** In-cell proximity-dependent biotinylation of BET proteins and neosubstrates through AirID-CRBN. HuH7 and MM1.S cells expressing AGIA-AirID-CRBN were treated with DMSO or 0.1-μM PROTACs in the presence of 10-μM biotin and 5-μM MG132 for 6 h.

The experiment was repeated twice independently, with similar results. **f** Neosubstrate selectivity of PROTACs based on 6-position-modified lenalidomides. NTERA-2 cells were treated with DMSO, 0.1-μM Po, 0.1-μM Le, 0.1-μM lenalidomide derivatives, or 0.1-μM PROTACs for 24 h. The experiment was independently repeated thrice, with similar results. **g–h** Immunoblot analysis of BET proteins and neosubstrates. (**g**) HuH7 or (**h**) MM1.S cells were treated with DMSO or PROTACs for 24 h. The experiment was independently repeated thrice, with similar results. **i** Heatmap showing the ratio of BET proteins to neosubstrates in whole-proteome quantification. NTERA-2 and MM1.S cells were treated with DMSO or 0.3-μM PRO-TACs for 16 h; quantitative proteomics analysis was performed using a tandem mass tag (biological replicates, n = 3). All relative AS signals are expressed as the luminescence signal relative to the luminescence signal of DMSO, which is considered 1. Error bars denote standard deviations (independent experiments, n = 3). Source data are provided as a Source data file.

based interaction assay showed that all PROTACs formed ternary complex of CRBN−PROTACs−BRD2/BRD3 (Fig. 6c). The hook effect of F-P and Cl-P was the same dose range as Po-P and Le-P, but that of $F_3C$-P shifted about ten times higher dose (Fig. 6c). As expected, PROTACs induced the protein degradation of BRD2, BRD3, and BRD4 in NTERA-2 cells (Supplementary Fig. 9a). To quantitatively examine protein degradation of BET proteins by the PROTACs, we generated HEK293T cells stably expressing BRD3-HiBit. Luminescent signals of BRD3-HiBit revealed that F-P induced protein degradation at the same dose as that of Le-P, but $DC_{50}$ of Cl-P and $F_3C$-P was higher than that of Le-P (Fig. 6d). Consistent with this result, protein degradation levels of BET protein at low concentrations of Cl-P and $F_3C$-P were lower than those under Le-P and F-P treatment (Supplementary Fig. 9b). Then, we examined cellular PROTAC-dependent interaction between CRBN and BET proteins using BioID enzymes. STA-PDA using AirID-CRBN showed that the biotinylation level of BET proteins by $F_3C$-P was slightly lower than that of the other PROTACs (Fig. 6e). These results suggest that 6-position-modified lenalidomides can be used as CRBN binder, though it was a possibility that optimisations including linkers and cellular permeability were required for a higher degradation ability to target proteins.

We then investigated the neosubstrate selectivity of the PROTACs based on 6-position-modified lenalidomides. As expected, STA-PDA showed that the biotinylation level of SALL4 and PLZF by F-P, Cl-P, and $F_3C$-P was lower than that by Le-P in HuH7 cells (Fig. 6e), and Cl-P and $F_3C$-P scarcely biotinylated IKZF1 and IKZF3 in MM1.S cells (Fig. 6e). Consistent with the neosubstrate selectivity of 6-position-modified lenalidomides, the degradation level of SALL4 by F-P and Cl-P was lower than that of Le-P, and $F_3C$-P scarcely degraded SALL4 (Fig. 6f and Supplementary Fig. 9a). Furthermore, F-P, Cl-P and $F_3C$-P did not induce protein degradation of PLZF and CK1α in HEK293T and MDS-L cells (Supplementary Fig. 9c, d). These results were validated via dose-dependent experiments using HuH7 cells, which expressed both SALL4 and PLZF (Fig. 6g). Additionally, immunoblot analysis showed that the protein degradation levels of IKZF1 and IKZF3 by Cl-P and $F_3C$-P were lower than those by Po-P, Le-P, and F-P in MM1.S, H929, and U266 cells (Fig. 6h and Supplementary Fig. 9e). To analyse protein degradation globally and quantitatively, we performed 18-plex TMT labelling and MS analysis of NTERA-2 and MM1.S cells (Supplementary Fig. 9f and Supplementary Data 2−3). The MS analysis showed that protein levels of BET proteins were the same among all PROTACs (Fig. 6i and Supplementary Fig. 10a, b). Importantly, protein degradation levels of SALL4, IKZF1 and IKZF3 by F-P, Cl-P, and $F_3C$-P were lower than those by Po-P and Le-P (Fig. 6i and Supplementary Fig. 10a, b). Po-P also induced protein degradation of ZFP91 but Le-P, F-P, Cl-P, and $F_3C$-P did not (Fig. 6i and Supplementary Fig. 10a, b). Alternatively, F-P and Cl-P degraded RAB28 but Po-P, Le-P, and $F_3C$-P did not (Fig. 6i and Supplementary Fig. 10a, b). These results suggest that the PRO-TACs retain neosubstrate selectivity of 6-position-modified lenalidomides.

### Anti-proliferative effect of PROTACs based on 6-position-modified lenalidomides

Next, we investigated whether the PROTACs based on 6-position-modified lenalidomide showed an anti-proliferative effect on cultured cells. Expectedly, the PROTACs based on 6-position-modified lenalidomide showed an anti-proliferative effect on MM1.S and H929 cells in a dose-dependent manner (Fig. 7a). Because Po, Le, F-Le and Cl-Le are effective in MM cell lines (Fig. 3d and Supplementary Fig. 11a), it is predicted that Po-P, Le-P, F-P and Cl-P showed stronger anti-proliferative effects than $F_3C$-P due to the dual protein degradation of IKZF1/IKZF3 and BET proteins (Fig. 7a). Therefore, to evaluate lenalidomide derivatives as CRBN binders, we investigated protein degradation and anti-proliferative effects in non-haematological cancer cell lines. BET proteins have been attractive targets for treating

diverse cancers, including neuroblastoma[50,51]. In the neuroblastoma cell line IMR32, F-P, Cl-P, and $F_3C$-P degraded BET proteins at the same dose range as Le-P (Fig. 7b). In the CellTiter-Glo assay, F-P and Cl-P showed anti-proliferative effects at the same dose as Le-P (Fig. 7c) but 6-position-modified lenalidomides did not show anti-proliferative effect (Supplementary Fig. 11b). The maximum anti-proliferative effect of $F_3C$-P was similar to that of other PROTACs and higher than that of OTX-015 (Fig. 7c). However, the anti-proliferative effect of $F_3C$-P at low doses was lower than that of other PROTACs (Fig. 7c). Fur-thermore, in the pluripotent human embryonal carcinoma cell line NTERA-2 and colon cancer cell line HCT116, F-P, Cl-P, and $F_3C$-P induced degradation of BET proteins in the same dose range as that of Le-P (Fig. 7d and Supplementary Fig. 11c). In NTERA-2 cells, F-P showed an anti-proliferative effect at the same level as that of Le-P, but the anti-proliferative effect of Cl-P and $F_3C$-P at low concentrations was lower than that of other PROTACs (Supplementary Fig. 11d), though 6-position-modified lenalidomides did not show an anti-proliferative effect (Supplementary Fig. 11e). In HCT116 cells, anti-proliferative effects of F-P and Cl-P were observed at the same dose as that of Le-P (Fig. 7e) but 6-position-modified lenalidomides did not affect the cell viability (Supplementary Fig. 11f). An effectiveness of the PROTACs in each cell line was showed as $GI_{50}$ and $GI_{max}$ (Fig. 7f). $GI_{50}$ values revealed that Le-P and F-P had same anti-proliferative effects on all cell lines examined in this study (Fig. 7f). Consistent with the degradation of BET proteins (Fig. 6d), $GI_{50}$ values of Cl-P and $F_3C$-P were lower than those of Le-P and F-P (Fig. 7f). Importantly, $GI_{max}$ values of all PROTACs were higher than that of OTX-015 (Fig. 7f), and $GI_{max}$ values of $F_3C$-P to HCT116 cells was the highest among those of all PROTACs (Fig. 7f). These results indicate that 6-position-modified lenalidomides can be CRBN binders of PROTACs for selective TPD.

## Discussion

Lenalidomide (Le) and pomalidomide are small-molecule drugs used at a scale of 16 billion US dollars annually worldwide. In particular, Le is the leading IMiD, associated with a cost of approximately 13 billion US dollars annually because it is used for treating MM and 5q MDS. In this study, we revealed that 6-position modification of Le with a small substituent increases the selectivity towards IKZF1, IKZF3, and CK1α, which are involved in anti-MM and anti-5q MDS activity[5–7]. Importantly, F-Le strongly induced protein degradation of these therapeutic targets (Figs. 1f, 2e) and the degradation of SALL4 and PLZF, which are involved in thalidomide teratogenicity, was weak (Figs. 1e and 2f–g). In fact, F-Le was more effective than lenalidomide in MM and 5q MDS cell lines (Fig. 3d–f). Cl-Le did not induce protein degradation of SALL4 and PLZF, although it degraded IKZF1, IKZF3, and CK1α (Figs. 2e–g and 3a, b). Additionally, Cl-Le showed an anti-proliferative effect on MM and 5q MDS cell lines, though the effectiveness of Cl-Le was lower than that of Le and F-Le (Fig. 3d–f). These results strongly suggest that F-Le and Cl-Le can be more selective and effective lenalidomide derivatives for treating MM and 5q MDS. However, more detailed investigations, such as those for bioactivity and bioavailability, are required.

IMiDs demonstrated that protein degradation could be a potent drug mechanism of action and that undruggable proteins, including transcription factors, can be targeted. Owing to the remarkable clinical success of IMiDs, novel thalidomide derivatives, such as CC-220[52] and CC-92480[53], are actively being developed to improve the degradation of IKZF1 and IKZF3. However, no reported thalidomide derivatives avoid the protein degradation of teratogenic targets. Interestingly, the chemical structures of CC-220 and CC-92480 were modified with each substituent on the amino group at the 4-position on lenalidomide. Since CC-220 and CC-92480 show considerable potential for drastic protein degradation of IKZF1 and IKZF3 at low doses[52,53], 6-position modification on CC-220 or CC-92480 may be a more selective and effective thalidomide derivative for the treat-ment of MM. Therefore, our results provide a promising direction

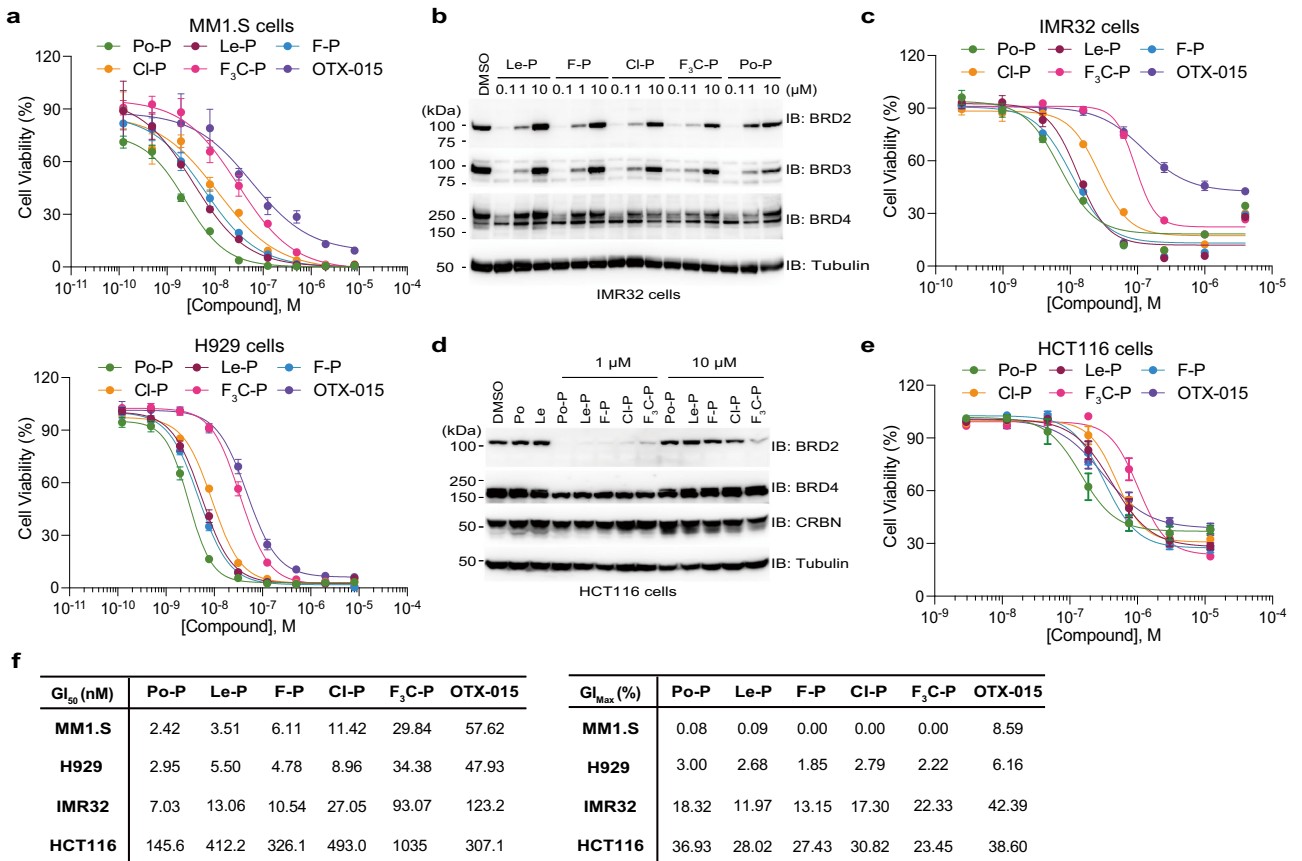

**Fig. 7 | Anti-proliferative effect of proteolysis-targeting chimeras (PROTACs) using 6-position-modified lenalidomides on diverse cancer cell lines.**
**a** Dose–response curves of anti-proliferative effects of PROTACs on MM cell lines. MM1.S and H929 cells were treated with dimethylsulfoxide (DMSO), PROTACs, or OTX-015 for 5 days, and cell viability was analysed using the CellTiter-Glo assay kit. Cell viability was expressed as the luminescence signal relative to the luminescence signal of DMSO, which was considered 100. Error bars denote standard deviation (biological replicates; $n = 4$). **b** Degradation of BET proteins by PROTACs in a neuroblastoma cell line. IMR32 cells were treated with DMSO or PROTACs for 24 h. The experiment was independently repeated thrice, with similar results.
**c** Dose–response curves of anti-proliferative effects of PROTACs on the neuroblastoma cell line. IMR32 cells were treated with DMSO, PROTAC, or OTX-015 for 4 days, and cell viability was analysed using the CellTiter-Glo assay kit. Cell viability was expressed as the luminescence signal relative to the luminescence signal of DMSO, which was considered 100. Error bars denote standard deviation (biological replicates; $n = 4$). **d** Degradation of BET proteins by PROTACs in a colon cancer cell line. HCT116 cells were treated with DMSO or PROTACs for 24 h. The experiment was independently repeated thrice, with similar results. **e** Dose–response curves of anti-proliferative effects of PROTACs on a colon cancer cell line. HCT116 cells were treated with DMSO, PROTACs, or OTX-015 for 4 days, and cell viability was analysed using the CellTiter-Glo assay kit. Cell viability was expressed as the luminescence signal relative to the luminescence signal of DMSO, which was considered 100. Error bars denote standard deviation (biological replicates; $n = 4$). **f** The half-maximal growth inhibition ($GI_{50}$) and the maximal growth inhibition ($GI_{max}$) values were calculated using dose–response curves in (**a, c, e**). Source data are provided as a Source data file.

for developing selective and effective thalidomide derivatives for the treatment of MM and 5q MDS.

In this study, we thoroughly investigated the change of neosubstrate selectivity caused by 6-position-modifications on Le (Figs. 4a–h and 5a–f). Br-Le scarcely degraded IKZF1, IKZF3, CK1α, SALL4, and PLZF (Fig. 4a–h). Additionally, F₃C-Le and F₃CO-Le did not completely degrade these neosubstrates (Fig. 4a–h). To understand the SARs, we analysed the docking model of 6-position-modified lenalidomides using crystal structures of IKZF1–CRBN[39], SALL4–CRBN[35], and CK1α-CRBN[33] complexes published in previous reports. It was predicted that 6-position-modified lenalidomides had almost the same binding mode as that of Le (Supplementary Fig. 4a) and all 6-position substituents were oriented towards the β-hairpin structure of each neosubstrate (Supplementary Fig. 4b). Because our biochemical analyses showed that all 6-position-modified lenalidomides had binding ability to CRBN (Supplementary Fig. 8a–f), the neosubstrate selectivity may depend on both the 6-position substituent and amino acids sequence of each neosubstrate. However, structural analyses are required to precisely understand the molecular basis. In the past decade, it was reported that many neosubstrates were degraded by IMiDs[9,39,41,42]. Furthermore,

many novel CRBN modulators targeting therapeutic targets, such as IKZF2 and GSPT1, are being developed[34,43,44]. Cell-based and proteomics analyses showed that Le, F-Le, Cl-Le, and Br-Le also degraded RAB28 in addition to IKZF1, IKZF3, CK1α, and SALL4 (Fig. 5b, c, f). F₃C-Le and F₃CO-Le scarcely induced protein degradation of reported neosubstrates (Fig. 5b, c, f). These results strengthen our argument that modifying the 6-position on lenalidomide with bulky molecules may disrupt neosubstrate degradation.

PROTACs are an alternative approach to inducing the degradation of target proteins[15,16]. Since the target binder does not need to be an inhibitor of target proteins, PROTACs enable us to target many types of proteins, including transcription factors, using many target binders[18–22]. Thalidomide derivatives are relatively small molecules among previously developed E3 binders[22]. The small molecular weight is a major advantage for developing PROTACs because a large molecular weight often causes problems in the cell permeability of the drug. Furthermore, it was reported that CRL4^CRBN was the most abundantly formed among the CRL4 complexes in cells[54]. However, it is a significant challenge to overcome the induction of degradation of neosubstrates by thalidomide derivative-based PROTACs[20]. Additionally,

thalidomide and its derivatives alter neosubstrate selectivity by metabolism, and the 5-hydroxyl metabolite selectively and strongly induces protein degradation of SALL4[10,35]. Although there is no report on whether the E3 binder generated as a metabolite of PROTACs induces protein degradation of neosubstrates, several studies have shown that the E3 binder is generated by metabolising PROTACs[55,56]. Therefore, improving the neosubstrate selectivity of the CRBN binder is crucial for developing more selective thalidomide derivative-based PROTACs in vivo. This study showed that PROTACs based on 6-position-modified lenalidomides had the same neosubstrate selectivity as that of the CRBN binders (Fig. 6e–i). However, a higher dose of F$_3$C-P-based PROTAC was required for the drastic degradation of target proteins and anti-proliferative effect on cell lines (Figs. 6d and 7a–f). The properties of PROTACs based on 6-position-modified lenalidomides should be analysed using PROTACs targeting other proteins in future studies. Previous studies have shown that linkers and attachment sites between the E3 and target binders are critical for the degradation of target proteins and neosubstrates[20,29,57]. Therefore, we believe that optimisations of the linker and attachment site may be required to develop more selective and effective PROTACs based on 6-position-modified lenalidomides.

In conclusion, our results provide crucial information for selective and effective TPD using thalidomide derivatives and thalidomide derivative-based PROTACs.

## Methods

### Reagents
Thalidomide (T2524, Tokyo Chemical Industry), pomalidomide (P2074, Tokyo Chemical Industry), lenalidomide (#126-06733, FUJI-FILM Wako Pure Chemical Corporation, Osaka, Japan), thalidomide-O-COOH (HY-103597, MedChemExpress, NJ, USA), CC-122/Avadomide, (HY-100507, MedChemExpress), CC-220/Iberdomide, (HY-101291, MedChemExpress), CC-885 (HY-101488, MedChemExpress), FPFT-2216 (HY-145319, MedChemExpress), biotin-thalidomide (#913979, Sigma-Aldrich, MA, USA), dBET1 (HY-101838, MedChemExpress), dBET6 (HY-112588, MedChemExpress), ARV-825 (HY-16954, MedChemExpress), MD-224 (HY-114312, MedChemExpress), ARV-110 (HY-138641, MedChemExpress), and MG132 (3175-v, Peptide Institute Inc., Osaka, Japan) were purchased. The procedure and physical data of thalidomide derivatives and PROTACs synthesised in this study are described in Supplementary Information. All drugs were dissolved in dimethylsulfoxide (DMSO, #045-24511; FUJIFILM Wako Pure Chemical Corporation) at 10–200 mM, stored at −30 °C as stock solutions, and diluted 1,000-fold for in vivo experiments or 100–200-fold for in vitro experiments.

### Plasmids
The pDONR221 and pcDNA3.1(+) plasmids were purchased from Invitrogen. The pEU plasmid for wheat cell-free protein synthesis was constructed in our laboratory[58]. The pcDNA3.1(+)-FLAG-GW, pcDNA3.1(+)-Myc-MCS, pcDNA3.1(+)-AGIA-MCS, pEU-bls-GW, pEU-FLAG-GST-GW and pEU-FLAG-GST-MCS plasmids were constructed by polymerase chain reaction (PCR) using the In-Fusion system (Takara Bio, #639648) or PCR and restriction enzymes. The pEU-FLAG-GST-SALL4, -IKZF1, -PLZF, -BRD2 and -BRD3 plasmids were purchased from the Kazusa DNA Research Institute[59]. AirID was purchased as an artificial gene from Thermo Fisher Scientific[38]. The open reading frames (ORFs) of *SALL4* and *IKZF1* were amplified, and restriction enzyme sites were added by PCR and cloned into pcDNA3.1(+)-AGIA-MCS or pcDNA3.1(+)-Myc-MCS. The ORF of *CRBN* was purchased from the Mammalian Gene Collection (MGC). The BP reaction sequence (attB and attP) was added to CRBN or the C-terminal domain of CRBN (residues 318–442) by PCR and cloned into pDONR221 using BP recombination (Thermo Fisher Scientific). Then, pDONR221-CRBN was recombined into pEU-bls-GW, pEU-FLAG-GST-GW or pcDNA3.1(+)-

FLAG-GW using LR recombination (attL and attR). The pcDNA3-3× HA-ubiquitin plasmid was kindly provided by Dr. Atsuo T. Sasaki (University of Cincinnati College of Medicine). For the generation of lentivirus for stable cell lines, AGIA-AirID-CRBN was cloned into the pCSII-CMV-MCS-IRES2-Bsd vector using restriction enzymes[30].

### Cell culture and transfection
HEK293T (#CRL-3216, American Type Culture Collection [ATCC]), HCT116 (#RCB2979, RIKEN BioResource Research Center [RBC], Kyoto, Japan), and MCF-7 (#JCRB0134, Japanese Collection of Research Bioresources [JCRB] cell bank) cells were cultured in low-glucose Dulbecco's modified Eagle's medium (DMEM, #041-29775; FUJIFILM Wako Pure Chemical Corporation) supplemented with 10% foetal bovine serum (#535-94155, FUJIFILM Wako Pure Chemical Corporation), 100 U/mL penicillin, and 100 µg/mL streptomycin (#15140122, Thermo Fisher Scientific, MA, USA) at 37 °C under 5% CO$_2$. HEK293T cells were transiently transfected using polyethyleneimine (PEI) Max (MW 40,000) (#2476, PolySciences Inc., PA, USA). HuH7 (#JCRB0403, JCRB cell bank) and HaCaT (#300493, CLS Cell Lines Service) cells were cultured in DMEM (high-glucose) medium (#044-29765, FUJIFILM Wako Pure Chemical Corporation) supplemented with 10% foetal bovine serum, 100 U/mL penicillin, 100 µg/mL streptomycin, 1 mM sodium pyruvate (#11360070, Thermo Fisher Scientific), 10 mM HEPES (#15630080, Thermo Fisher Scientific), and 1× MEM NEAA (#11140050, Thermo Fisher Scientific) at 37 °C under 5% CO$_2$. MM1.S (#CRL-2974, ATCC), H929 (#95050415, European Collection of Authenticated Cell Cultures [ECACC]), U266 (#TIB-196, ATCC), RPMI8226 (#JCRB0034, JCRB cell bank), SKM-1 (#JCRB0118, JCRB cell bank), KG-1 (#JCRB0065, JCRB cell bank), Jurkat (#TIB-152, ATCC), TK (#JCRB0157, JCRB cell bank), BJAB (#ACC757, Deutsche Sammlung von Mikroorganismen und Zellkulturen [DSMZ]), Karpas-1106P (#06072607, ECACC), SU-DHL-4 (#CRL2957, ATCC) and THP-1 (#RCB3686, RIKEN RBC) cells were cultured in RPMI 1640 GlutaMAX medium (#72400047, Thermo Fisher Scientific) supplemented with 10% foetal bovine serum, 100 U/mL penicillin, 100 µg/mL streptomycin, and 55 µM 2-mercaptoethanol (#21985023, Thermo Fisher Scientific) at 37 °C under 5% CO$_2$. KG-1a (#RCB1928, RIKEN RBC) cells were cultured in RPMI 1640 GlutaMAX medium supplemented with 20% foetal bovine serum, 100 U/mL penicillin, 100 µg/mL streptomycin, and 55 µM 2-mercaptoethanol at 37 °C under 5% CO$_2$. MDS-L cells (originally prepared and provided by Prof. K. Tohyama[60] [Kawasaki Medical School]) were cultured in RPMI 1640 medium (#189-02145, FUJIFILM Wako Pure Chemical Corporation) supplemented with 20% foetal bovine serum, 100 U/mL penicillin, 100 µg/mL streptomycin, 20 ng/mL IL-3 (#578002, BioLegend, CA, USA), and 55 µM 2-mercaptoethanol at 37 °C under 5% CO$_2$. NTERA-2 (#01071221, ECACC) cells were cultured in DMEM (high glucose) (#11965092, Thermo Fisher Scientific) supplemented with 10% foetal bovine serum, 100 U/mL penicillin, 100 µg/mL streptomycin, 2 mM L-glutamine (#A2916801, Thermo Fisher Scientific) and 1× MEM NEAA at 37 °C under 5% CO$_2$. IMR32 (#JCRB9050, JCRB cell bank) cells were cultured in MEM GlutaMAX medium (#41090036, Thermo Fisher Scientific) supplemented with 10% fetal bovine serum, 100 U/mL penicillin, 100 µg/mL streptomycin, and 1× MEM NEAA at 37 °C under 5% CO$_2$.

For the generation of a cell line stably expressing AGIA-AirID-CRBN, lentivirus was produced in HEK293T cells by transfection of pCSII-CMV-AirID-CRBN-IRES2-Bsd expression vector together with pCMV-VSV-G-RSV-Rev and pCAG-HIVgp as described in previous study[30]. HuH7 and MM1.S cells were infected with the lentivirus and selected using blasticidin S (#029-18701, FUJIFILM Wako Pure Chemical Corporation) in a previous study[30].

For the generation of a cell line stably expressing IKZF1-HiBiT, CK1α-HiBiT, SALL4-HiBiT or PLZF-HiBiT, lentivirus was produced in HEK293T cells by transfection of pCSII-CMV-neosubstrate-HiBiT-IRES2-Bsd expression vector together with pCMV-VSV-G-RSV-Rev and

pCAG-HIVgp as described in previous study[30]. HEK293T cells were infected with the lentivirus and selected using blasticidin S in a previous study[30].

For the generation of a cell line stably expressing BRD3-HiBiT, lentivirus was produced in HEK293T cells by transfection of pLVSIN-EF1α-BRD3-HiBiT-Pur (#6186, Takara Bio, Shiga, Japan) using Lentiviral High Titer Packaging Mix (#6194, Takara Bio). HEK293T cells supplemented with 10 μg/mL polybrene (#12996-81, Nacalai Tesque Inc., Kyoto, Japan) were infected with the lentivirus. The infected cells were cultured for 24 h, and the fresh culture medium was added. A 1 μg/mL puromycin (#P9620, Sigma-Aldrich) selection was started 24 h after culture medium exchange.

## Antibodies

The following horseradish peroxidase (HRP)-conjugated antibodies were used: FLAG (Sigma-Aldrich, #A8592, 1:5000), AGIA[61] (produced in our laboratory, 1:5000), Myc-tag (Cell Signaling Technology, #2040, 1:1000), HA-tag (Roche, #12013819001, 1:5000), α-tubulin (MBL, #PM054-7, 1:5000), GAPDH (MBL, #M171-7, 1:5000), streptavidin (Abcam, ab7403, 1:5000) and biotin (Cell Signaling Technology, #7075, 1:3000). The following primary antibodies were used: CRBN (#71810, 1:1000), IKZF1/Ikaros (#14859, 1:1000), IKZF2/Helios (#42427, 1:1000), IKZF3/Aiolos (#15103, 1:1000), IRF4 (#62834, 1:1000), c-Myc (#18583, 1:1000), RUNX1 (#4336, 1:1000), BRD4 (#13440, 1:1000), GSPT1 (#14980, 1:1000), p63-α (#13109, 1:1000), GAPDH (#5174, 1:1000), (all from Cell Signaling Technology); BRD4 (#A301-985A, 1:1000), BRD2 (#A302-583A, 1:1000), ZFP91 (#A303-245A, 1:1000) (all from Bethyl Laboratories); PLZF (R&D System, #AF2944, 1:1000); SALL4 (#sc-101147, 1:500), BRD3 (#sc-81202, 1:500), MEIS2 (#sc-81986, 1:500) (all from Santa Cruz Biotechnology); CK1α (Abcam, #ab108296, 1:1000); IKZF4/Eos (GeneTex, #GTX128043, 1:1000); ZMYM2 (Gene-Tex, #GTX105550, 1:1000); RAB28 (ABclonal, #A17368, 1:500); RNF166 (ABclonal, #A8276, 1:500); and α-tubulin (LI-COR Biosciences, #926-42213, 1:1000). Anti-rabbit IgG (HRP-conjugated, Cell Signaling Technology, #7074, 1:5000), anti-mouse IgG (HRP-conjugated, Cell Signaling Technology, #7076, 1:5000), anti-goat IgG (HRP-conjugated, Thermo Fisher Scientific, #81-1620, 1:10000), IRDye 800CW goat anti-rabbit IgG (LI-COR Biosciences, #925-32211, 1:10000), IRDye 680RD goat anti-mouse IgG (LI-COR Biosciences, #925-68070, 1:10000), IRDye 800CW goat anti-mouse IgG (LI-COR Biosciences, #925-32210, 1:10000), and IRDye 680RD goat anti-rabbit IgG (LI-COR Biosciences, #925-68071, 1:10000) were used as secondary antibodies.

## Immunoblot analysis

Each cell pellet was lysed in RIPA buffer (25 mM Tris-HCl pH 8.0, 150 mM NaCl, 1% NP-40, 0.5% sodium deoxycholate, 0.1% sodium dodecyl sulphate (SDS), 1 mM EDTA) containing protease inhibitor cocktail (#P8340, Sigma-Aldrich). The cell lysates were centrifuged at 16,100 × g for 15 min, and protein concentration in the supernatant was quantified by BCA assay kit (#23227, Thermo Fisher Scientific). Then, the lysates were denatured by boiling in 1× sample buffer (62.5 mM Tris-HCl pH 6.8, 2% SDS, 10% glycerol) containing 5% 2-mercaptoethanol. The equal amount of lysate was separated by SDS-PAGE and transferred to polyvinylidene difluoride (PVDF) membranes (#IPVH00010, Millipore). The membranes were blocked using 5% skim milk (Megmilk Snow Brand) in TBST (20 mM Tris-HCl [pH 7.5], 150 mM NaCl, 0.05% Tween20) or Intercept (TBS) Blocking Buffer/TBS (#927-60001, LI-COR Biosciences) at room temperature for 1 h and incubated with primary antibodies overnight at 4 °C. Then, the membranes were washed using TBST for 15 min and incubated with secondary antibodies at room temperature for 1 h. ImmunoStar LD (#290-69904, FUJIFILM Wako Pure Chemical Corporation) or EzWestLumi plus (#2332638, Atto, Korea) were used as a substrate for HRP, and the luminescence signal was detected using an ImageQuant LAS 4000 mini (GE Healthcare, version 1.1). For quantification of

immunoblot analyses of MEIS2, band intensity was measured using Image J (Fiji) (version 2.1.0). In some blots, the membrane was stripped with a stripping solution (#193-16375, FUJIFILM Wako Pure Chemical Corporation) and re-probed with other antibodies. For fluorescent immunoblot analysis, the fluorescent signal was detected using an Odyssey Fc (LI-COR Biosciences, version 5.2) and analysed using Empiria Studio software (version 1.3).

## Production of recombinant proteins using a cell-free system

Recombinant protein synthesis was conducted using a wheat cell-free system. In vitro transcription and wheat cell-free protein synthesis were performed using the WEPRO1240 expression kit (#CFS-TRI-1240, Cell-Free Sciences, Ehime, Japan). Transcription was performed using SP6 RNA polymerase with the plasmids or DNA fragments as templates. The translation reaction was performed in the bilayer mode using the WEPRO1240 expression kit (#CFS-TRI-1240, Cell-Free Sciences), according to the manufacturer's protocol. For biotin labelling of bls-CRBN, cell-free synthesised crude biotin ligase (BirA) produced using the wheat cell-free expression system was added to the bottom layer, and 0.5 μM (final concentration) of d-biotin (#04822-91, Nacalai Tesque) was added to both the upper and lower layers, as described previously[62]. For the production and purification of recombinant SALL4 and IKZF1, FLAG-GST-SALL4 or -IKZF1 was synthesised on a 6 mL scale using the WEPRO1240G expression kit (#CFS-TRI-1240G, Cell-Free Sciences). Then, the crude protein solutions containing NaCl (final concentration 100 mM) and DTT (final concentration 10 mM) were rotated with 200 μL MagneGST Glutathione Particles (Promega) at 4 °C for 3 h. The beads were washed three times with 800 μL phosphate-buffered saline (PBS) and incubated two times with 150 μL elution buffer (50 mM Tris-HCl [pH 8.0], 100 mM NaCl, 10 mM reduced glutathione) on ice for 15 min. The purified proteins were confirmed using Coomassie brilliant blue (CBB)-staining, and the protein concentration was calculated by the band intensity of each purified protein using Image J (Fiji) (version 2.1.0).

## AlphaScreen-based biochemical assays using recombinant proteins

In vitro biochemical interaction assay was performed as described previously[10]. 10 μL CRBN mixtures containing 0.5 μL biotinylated bls-CRBN in AlphaScreen buffer (100 mM Tris [pH 8.0], 0.01% Tween20, 100 mM NaCl, and 1 mg/mL bovine serum albumin [BSA]) were prepared. 5 μL compound mixtures containing thalidomide derivatives were prepared in AlphaScreen buffer. 5 μL substrate mixtures containing 0.8 μL FLAG-GST-SALL4, -IKZF1 or -PLZF in AlphaScreen buffer were prepared. Then, the three mixtures were dispensed and incubated at 26 °C for 1 h in a 384-well AlphaPlate (PerkinElmer). Subsequently, a 5 μL detection mixture containing 0.2 μg/mL anti-DYKDDDDK mouse mAb (#012-22384, FUJIFILM Wako Pure Chemical Corporation), 0.08 μL streptavidin-coated donor beads (#6760617, PerkinElmer, MA, USA), and 0.08 μL Protein A-coated acceptor beads (#6760617, PerkinElmer) in AlphaScreen buffer were added to each well and incubated. After incubation at 26 °C for 1 h, luminescence signals were detected using an EnVision plate reader (PerkinElmer version 1.12).

## In vitro pull-down assay of CRBN and neosubstrate

Ten microlitres biotinylated bls-CRBN and 10 μL Dynabeads MyOne Streptavidin C1 (DB65002, VERITAS) were mixed and rotated at 26 °C for 1 h. The beads were washed three times with 500 μL PBS containing 0.05% Tween20, and then 300 μL reaction solutions containing 20 μL FLAG-GST-SALL4 or -IKZF1 and DMSO or 100 μM thalidomide derivatives (0.5% DMSO) in AlphaScreen buffer were added. After incubation at room temperature for 90 min, the beads were washed three times with 500 μL IP Lysis buffer (Pierce) (25 mM Tris-HCl pH [7.5], 150 mM NaCl, 1 mM EDTA, 1% NP-40, and 5% glycerol). The proteins were eluted

by boiling with 1× sample buffer containing 5% 2-mercaptoethanol and analysed using immunoblotting.

## In vitro ubiquitination assays

FLAG-GST-SALL4 and -IKZF1 were obtained by protein synthesis and purification using the wheat cell-free system described above. The recombinant CRL4$^{CRBN}$ complex was purchased from the R&D system (E3-650). Then, 100 nM FLAG-GST-SALL4 or -IKZF1 in 30 μL 1× ubiquitination reaction buffer (20 mM HEPES pH [7.5], 150 mM NaCl, and 10 mM MgCl$_2$) containing 200 nM UBE1 E1 (R&D systems, E-305), 1 μM UbcH5a E2 (Enzo, BML-UW9050-100), 10 μM HA-ubiquitin (R&D systems, U-110) and DMSO or 20 μM thalidomide derivatives (1% DMSO) was incubated at 30 °C for 30 min. Then, 100 mM ATP (final concentration 5 mM) was mixed, and in vitro ubiquitin reaction was performed at 30 °C for 3 h. The proteins were denatured in 1% SDS by boiling at 95 °C for 15 min. The proteins were diluted 10-fold with IP Lysis buffer (Pierce) and immunoprecipitated with anti-FLAG M2 magnetic beads (#M8823, Sigma-Aldrich) at 4 °C for 4 h. The beads were washed three times with 800 μL IP Lysis buffer (Pierce), and the proteins were eluted by boiling with 20 μL 1× sample buffer containing 5% 2-mercaptoethanol and analysed using immunoblotting.

## In cell ubiquitination assays

HEK293T cells were cultured in a 6-well plate and transfected with 500 ng pcDNA3.1( + )- FLAG-CRBN, 500 ng pcDNA3.1(+)-AGIA-SALL4 or -IKZF1 and 400 ng pcDNA3 3× HA-ubiquitin. After 16 h of incubation from transfection, the cells were treated with DMSO, 1 μM pomalidomide, 10 μM lenalidomide or 10 μM lenalidomide derivatives in the presence of 10 μM MG132 for 8 h. The cells were lysed in 150 μL of SDS lysis buffer (50 mM Tris−HCl pH [7.5], 1% SDS) containing a protease inhibitor cocktail (Sigma-Aldrich) and denatured at 90 °C for 15 min. The lysates were treated with Benzonase Nuclease (#E1014, Sigma-Aldrich) at 37 °C for 30 min, and 120 μL the lysates were centrifuged at 16,100 × g for 15 min and then diluted 10-fold with IP Lysis buffer (Pierce). The proteins were immunoprecipitated overnight with Dynabeads Protein G (DB10004, VERITAS) interacting anti-AGIA antibody at 4 °C, which were then washed three times with 800 μL of IP Lysis buffer (Pierce). Proteins were eluted by boiling in 25 μL 1× sample buffer containing 5% 2-mercaptoethanol. The proteins were then analysed by immunoblot.

## In vitro competition assay using thalidomide derivatives

The 4 mM thalidomide-immobilised magnetic beads were generated using thalidomide-O-COOH (HY-103597, MedChemExpress) and FG-beads (TAS8848N1130, Tamagawa Seili Co., Ltd., Tokyo, Japan) according to the manufacturer's instruction. Then, 10 μL FLAG-GST-CRBN and 10 μL 4 mM thalidomide-immobilised magnetic beads in 500 μL IP Lysis buffer (Pierce) were rotated at room temperature for 2 h. The beads were washed four times with 800 μL IP Lysis buffer (Pierce) and eluted with 20 μL IP Lysis buffer (Pierce) containing DMSO or 200 μM thalidomide derivatives (1% DMSO) by vortexing at 26 °C for 30 min. The eluted proteins were denatured by boiling with 1× sample buffer containing 5% 2-mercaptoethanol and analysed using immunoblotting.

For competitive assay using AlphaScreen technology, 15 μL CRBN−thalidomide mixtures containing 0.5 μL FLAG-GST-CRBN (318–426) and biotinylated thalidomide (final concentration 50 nM) in AlphaScreen buffer (100 mM Tris [pH 8.0], 0.01% Tween20, 100 mM NaCl, and 1 mg/mL BSA) were prepared. Then, 5 μL compound mixtures containing thalidomide derivatives or PROTACs were prepared in AlphaScreen buffer. The two mixtures were dispensed and incubated at 26 °C for 1 h in a 384-well AlphaPlate (PerkinElmer). Subsequently, 5 μL detection mixture containing 0.2 μg/mL anti-DYKDDDDK mouse mAb, 0.08 μL streptavidin-coated donor beads, and 0.08 μL Protein A-coated acceptor beads in AlphaScreen buffer was added to

each well and incubated. After incubation at 26 °C for 1 h, luminescence signals were detected using an EnVision plate reader (PerkinElmer).

## Docking simulation

Three crystal structures of IKZF1-pomalidomide-CRBN (PDB ID: 6H0F)[39], SALL4-thalidomide-CRBN (PDB ID: 7BQU)[35] and CK1α-lenalidomide-CRBN complexes (PDB ID: 5FQD)[33] were selected for the docking simulation of 6-position modified lenalidomide. Polar hydrogens and charges were added to the thalidomide-binding domain (TBD, 318−427 residues) of CRBN and the full-length of neo-substrates extracted from each coordination file using AutoDockTools-1.5.6 (The Scripts Research Institute, version 1.5.6). 3D models of 6-position modified lenalidomides were created using the PRODRG sever[63]. After fine-tuning the orientation of the phthalimide ring of each derivative to match that of the lenalidomide molecule bound to the CK1α-CRBN complex using PyMOL Molecular Graphics System (version 2.4.0 Schrödinger, LLC), UCSF Chimera 1.14[64] was used to add hydrogen and charge to lenalidomide and all the derivatives, and to minimise the models energetically. IMiD-binding sites were selected as docking sites, and the grid box was set to wrap around the site for each protein model. Some residues were set to adopt variable side-chain conformation: H353 for CRBN, Q146 and N148 for IKZF1, V411 and S413 for SALL4, and K18, I35 and I37 for CK1α. Docking simulation was performed using AutoDock Vina[40] with the following parameters: exhaustiveness, 8; number of modes, 100; and energy range, 3. First, we confirmed that the docking simulation worked well using lenalidomide as a ligand. Next, 6-position modified lenalidomide was applied to generate the docking models, which were automatically ranked based on the calculated affinity score (kcal mol$^{-1}$) by AutoDock Vina. The PyMOL Molecular Graphics System was used to depict all the structures.

## CRBN TBD expression and purification

For ITC measurements, DNA sequences encoding human CRBN TBD (318–426 and C366S mutation) were cloned into pGEX6P-3 (GE Healthcare), and the recombinant CRBN TBD was expressed in E. coli Rossetta (DE3) cells (Novagen) using lysogeny-broth (LB) media supplemented with 100 μg mL$^{-1}$ ampicillin and 17 μg mL$^{-1}$ chloramphenicol. Protein expression was induced by adding 0.5 mM isopropyl β-D-thiogalactopyranoside at 18 °C when the optical density at 600 nm (OD600) reached -0.6. The cells were collected by centrifugation and resuspended in a buffer containing 20 mM Tris-HCl, pH 8.0, 500 mM NaCl, and 0.1 mM tris(2-carboxyethyl)phosphine (TCEP). After sonication and centrifugation, the supernatant of the cell lysate was passed over glutathione Sepharose 4B resin (GE Healthcare), and resin-bound proteins were cleaved overnight using human rhinovirus 3 C (HRV3C) protease. Proteins eluted from the resin were purified using size-exclusion chromatography with Superdex 75 10/300 GL (GE Healthcare) in 50 mM sodium phosphate, pH 7.4, 200 mM NaCl, and 0.1 mM TCEP. The fractions containing the CRBN TBD were pooled and concentrated by ultrafiltration with Vivaspin 20 (MWCO 3000, Sartorius). The concentration of the CRBN TBD was determined by measuring the ultraviolet absorbance at 280 nm, and the molecular extinction coefficient was 27,960 M$^{-1}$ cm$^{-1}$.

## ITC measurements

The binding affinity of (R/S)-lenalidomide, (R/S)-F-lenalidomide, (R/S)-Cl-lenalidomide, (R/S)-Br-lenalidomide, (R/S)-F$_3$C-lenalidomide and (R/S)-F$_3$CO-lenalidomide to the CRBN TBD was measured by using an isothermal titration calorimeter (MicroCal iTC$_{200}$, Malvern). The CRBN TBD was dialysed in a binding buffer containing 50 mM sodium phosphate, pH 7.4, 200 mM NaCl, and 0.1 mM TCEP, and then DMSO was added to the protein solution at a final concentration of 0.2%. (R/S)-lenalidomide, (R/S)-F-lenalidomide, (R/S)-Cl-lenalidomide, (R/S)-

Br-lenalidomide, (*R/S*)-F$_3$C-lenalidomide or (*R/S*)-F$_3$CO-lenalidomide was dissolved in DMSO, and the solution was mixed with binding buffer with the DMSO concentration adjusted to 0.2%. For titrations, the (*R/S*)-lenalidomide solution (400 μM, in the syringe), (*R/S*)-F-lenalidomide solution (400 μM, in the syringe), (*R/S*)-Cl-lenalidomide solution (400 μM, in the syringe), (*R/S*)-Br-lenalidomide solution (400 μM, in the syringe), (*R/S*)-F$_3$C-lenalidomide solution (400 μM, in the syringe) or (*R/S*)-F$_3$CO-lenalidomide solution (300 μM, in the syringe) was injected into the sample cell filled with the CRBN TBD solution (10 μM or 20 μM, in the cell) in 37 consecutive 1.0 μL aliquots at 120 s intervals. The first injection volume was 0.4 μl, and the observed thermal peak was excluded from the data analyses. All experiments were performed at 25 °C with a reference power of 5 μcal sec$^{-1}$ and a stirring speed of 750 rpm. Data fitting was performed using Origin 7.0 software (OriginLab) in the "one set of sites" mode. The values of the dissociation constant ($K_D$) and molar binding ratio ($n$) for each lenalidomide-derivatives were calculated from the data obtained in triplicate experiments (means ± SD).

### STA-PDA using BioID

STA-PDAs using AirID-CRBN were performed as described previously[30]. MM1.S or HuH7 cells stably expressing AGIA-AirID-CRBN were cultured in a 6-well plate and treated with DMSO or 10 μM thalidomide derivatives in the presence of 10 μM biotin and 5 μM MG132 for 6 h. The cells were harvested using a cell scraper and lysed in 300 μL SDS lysis buffer (50 mM Tris-HCl, pH 7.5, and 1% SDS) containing a protease inhibitor cocktail (Sigma-Aldrich), and then the lysates were denatured by boiling at 95 °C for 15 min. The lysates were treated with Benzonase Nuclease (Sigma-Aldrich) at 37 °C for 30 min and centrifuged at 16,100 × *g* for 15 min. Subsequently, 250 μL lysates were added to 1 mL IP Lysis buffer (Pierce) containing 20 μL Dynabeads MyOne Streptavidin C1 and rotated at 4 °C overnight. The beads were washed three times with 600 μL IP Lysis buffer (Pierce), and the proteins were eluted by boiling with 40 μL 2 × sample buffer containing 5% 2-mercaptoethanol.

For the STA-PDA using PROTAC, MM1.S or HuH7 cells stably expressing AGIA-AirID-CRBN were cultured in a 10 cm dish and treated with DMSO or 100 nM PROTAC in the presence of 10 μM biotin and 5 μM MG132 for 6 h. The cells were harvested using a cell scraper and lysed in 600 μL SDS lysis buffer (50 mM Tris-HCl, pH 7.5, and 1% SDS) containing a protease inhibitor cocktail (Sigma-Aldrich), and then the lysates were denatured by boiling at 95 °C for 15 min. The lysates were treated with Benzonase Nuclease (Sigma-Aldrich) at 37 °C for 30 min and centrifuged at 16,100 × *g* for 15 min. Subsequently, 560 μL lysates were added to 1 mL IP Lysis buffer (Pierce) containing 25 μL Dynabeads MyOne Streptavidin C1 and rotated at 4 °C overnight. The beads were washed three times with 800 μL IP Lysis buffer (Pierce), and the proteins were eluted by boiling with 40 μL 2 × sample buffer containing 5% 2-mercaptoethanol.

### Quantitative degradation assay using HiBiT system

For analysis of protein degradation of IKZF1, CK1α, SALL4 and PLZF, HEK293T cells stably expressing IKZF1-HiBiT, CK1α-HiBiT, SALL4-HiBiT, or PLZF-HiBiT were cultured in 96-well plates and treated with DMSO or thalidomide derivatives for 24 h. Then, the cells were lysed using Nano-Glo HiBiT Lytic Detection System (N3040, Promega) according to the manufacturer's instruction. The luminescence signals of HiBiT-tagged neosubstrates were detected using SpectraMax iD3 (Molecular Devices).

For analysis of protein degradation of BRD3 cells stably expressing BRD3-HiBiT, they were cultured in 96-well plates and treated with DMSO or PROTACs for 6 h. Then, the cells were lysed using Nano-Glo HiBiT Lytic Detection System according to the manufacturer's instruction. The luminescence signal of the HiBiT-tagged neosubstrate was detected using SpectraMax iD3 (Molecular Devices).

### Quantitative RT-PCR

MDS-L cells were cultured in 48-well plates and treated with DMSO, 10 μM pomalidomide, 10 μM lenalidomide or 10 μM lenalidomide derivatives for 3 days. Then, half of the cells were collected into a tube, and the remaining cells were diluted 2-fold with a culture medium. The diluted cells were cultured for up to 6 days and collected in a tube. All collected cells were lysis using SuperPrep II Cell Lysis Kit for qPCR (SCQ-501, Toyobo Co., Ltd., Osaka, Japan), and total RNA was isolated using SuperPrep II Cell Lysis Kit for qPCR, according to the manufacturer's instructions. RT-PCR was performed using KOD SYBR qPCR Mix (QKD-201, Toyobo), and the data were normalised against GAPDH mRNA levels. PCR primers were as follows: *SELP* sense 5´- TCCGCT GCATTGACTCTGGACA −3´, *SELP* anti-sense 5´- CTGAAACGCTCTCAA GGATGGAG-3´, *ITGB3* sense 5´-CATGGATTCCAGCAATGTCCTCC-3´, *ITGB3* antisense 5´- TTGAGGCAGGTGGCATTGAAGG-3´, *GAPDH* sense 5´-AGCAACAGGGTGGTGGAC-3´, and *GAPDH* antisense 5´- GTGTGGT GGGGGACTGAG-3´.

### Cell viability assay

To evaluate the anti-proliferative effect of thalidomide derivatives on multiple myeloma and 5q myelodysplastic syndrome cell lines, MM1.S, U266, H929, RPMI8226, KG-1, KG-1a, SKM-1, or MDS-L cells were cultured in 24-well plates in the presence of DMSO, thalidomide, pomalidomide, lenalidomide, or lenalidomide derivatives at the indicated concentrations. The cells were cultured for 5 days or diluted 4-fold every 3 days and cultured for 9 days. The cells were lysed using a Cell-Titer-Glo assay kit (G7572, Promega) and dispensed in a 384-well OptiPlate (PerkinElmer). Luminescent signals were detected using SpectraMax iD3 (Molecular Devices, version 7.1) according to the manufacturer's instructions. To evaluate anti-proliferative effect of lenalidomide derivatives or PROTACs on diverse cultured-cells, MM1.S, H929, IMR32, NTERA-2, or HCT116 cells were cultured in 96-well plates in the presence of DMSO, thalidomide, pomalidomide, lenalidomide, lenalidomide derivatives, or PROTACs at indicated concentrations for 5 days. The cells were lysed using the Cell-Titer-Glo assay kit and dispensed in a 384-well OptiPlate (PerkinElmer). Luminescent signals were detected using SpectraMax iD3 (Molecular Devices) according to the manufacturer's instructions.

For the dose–response curve of the anti-proliferative effect of 6-position-modified lenalidomides on MM cell lines, MM1.S or H929 cells were cultured in 96-well plates treated with DMSO, pomalidomide, lenalidomide, or 6-position-modified lenalidomides every 5 days for 10 days. For the dose–response curve of the anti-proliferative effect of 6-position-modified lenalidomides on a 5q MDS cell line, MDS-L cells were cultured in 24-well plates treated with DMSO, pomalidomide, lenalidomide, or 6-position-modified lenalidomides every 2 days for 12 days. For the dose–response curve of the anti-proliferative effect of PROTACs based on 6-position-modified lenalidomides on diverse cultured cells, MM1.S, H929, IMR32, or HCT116 cells were cultured in 96-well plates and treated with the PROTACs for 4 or 5 days. The cells were lysed using the Cell-Titer-Glo assay kit and dispensed in a 384-well OptiPlate (PerkinElmer). Luminescent signals were detected using SpectraMax iD3 (Molecular Devices) according to the manufacturer's instructions.

### TMT-based quantitative proteomics

For global investigation of protein degradation by 6-position-modified lenalidomides, MM1.S cells were cultured in 6-well plates and treated with DMSO or 10 μM thalidomide or its derivatives for 5 h. The cells were harvested by suspending and centrifuged at 400 × g for 3 min, after which the cell pellets were washed with PBS. Then, the cell pellets were lysed in 200 μL guanidine buffer (6 M guanidine-HCl, 100 mM HEPES-NaOH, pH 7.5, 10 mM TCEP, and 40 mM chloroacetamide). For global investigation of protein degradation by PROTACs, MM1.S or NTERA-2 cells were cultured in 6-well plates and treated with DMSO or

300 nM PROTACs for 16 h. MM1.S or NTERA-2 cells were harvested by suspending or scraping and centrifuged at 400 × g for 3 min, after which the cell pellets were washed with PBS. Then, the cell pellets were lysed in 100 μL guanidine buffer. After heating and sonication, proteins (100 μg each) were purified using methanol–chloroform precipitation and resuspended in 20 μL 0.1% RapiGest SF (Waters) in 50 mM triethylammonium bicarbonate. After sonication and heating at 95 °C for 10 min, the proteins were digested with 2 μg trypsin/Lys-C mix (V5072, Promega) at 37 °C overnight. The digested peptides (40 μg each) were labelled with 0.5 mg TMTpro-18plex reagents (A52045, Thermo Fisher Scientific) for 1 h at 25 °C. After the reaction was quenched with hydroxylamine, all the TMT-labelled samples were pooled, acidified with trifluoroacetic acid (TFA), and fractionated using offline high-pH reversed-phase chromatography on a Vanquish DUO UHPLC system (Thermo Fisher Scientific), as previously reported with slight modifications[65]. Briefly, the peptides were loaded onto a 4.6 × 250 mm Xbridge BEH130 C18 column with 3.5 mm particles (Waters) and separated using a 30 min multistep gradient of solvents A (10 mM ammonium formate at pH 9.0 in 2% acetonitrile [ACN]) and B (10 mM ammonium formate pH 9.0 in 80% ACN), at a flow rate of 1 mL/min. Peptides were separated into 48 fractions, which were consolidated into 16 fractions. Each fraction was evaporated in a SpeedVac concentrator and dissolved in 0.1% TFA and 3% ACN. LC-MS/MS analysis of the resultant peptides (500 ng each) was performed on an EASY-nLC 1200 UHPLC system connected to a Q Exactive Plus mass spectrometer through a nanoelectrospray ion source (Thermo Fisher Scientific). The peptides were separated on the analytical column (75 μm × 15 cm, 3 μm; Nikkyo Technos) with a linear gradient of 4–20% ACN for 0–115 min and 20–32% ACN for 115–160 min, followed by an increase to 80% ACN for 10 min and finally held at 80% ACN for 10 min. The mass spectrometer was operated in data-dependent acquisition mode with a top 10 MS/MS method. MS1 spectra were measured with a resolution of 70,000, an AGC target of 3e6, and a mass range from 375 to 1,400 $m/z$. MS/MS spectra were triggered at a resolution of 35,000, an AGC target of 1e5, an isolation window of 0.7 $m/z$, a maximum injection time of 150 ms, and a normalised collision energy of 33. Dynamic exclusion was set to 20 s. Raw data were directly analysed against the SwissProt database restricted to *Homo sapiens* using Proteome Discoverer version 2.4 with the Sequest HT search engine for identification and TMT quantification. The search parameters were as follows: (a) trypsin as an enzyme with up to two missed cleavages; (b) precursor mass tolerance of 10 ppm; (c) fragment mass tolerance of 0.02 Da; (d) TMT of lysine and peptide N-terminus and carbamidomethylation of cysteine as fixed modifications; (e) oxidation of methionine as a variable modification. Peptides were filtered at an FDR of 1% using the Percolator node. TMT quantification was performed using the Reporter Ions Quantifier node. Normalisation was performed such that the total sum of the abundance values for each TMT channel over all peptides was the same.

## Statistical analysis and reproducibility
The data are presented as mean ± standard deviation (SD). Significant changes were analysed using Student's *t* tests or one-way analysis of variance (ANOVA), followed by Tukey's tests using Microsoft Excel (version 16.66) or GraphPad Prism (version 9) (GraphPad, Inc.). Each value determined from the dose–response curve was calculated using GraphPad Prism (version 9). All experiments were repeated more than twice, and the number of replications is described in the figure legends.

## Reporting summary
Further information on research design is available in the Nature Portfolio Reporting Summary linked to this article.

## Data availability
The MS proteomics data have been provided in Supplementary Data 1–3 and deposited to the ProteomeXchange Consortium via the jPOST partner repository with the dataset identifiers PXD041812 (TMTpro-18plex analysis of MM1.S cells treated with thalidomide and its derivatives), PXD037178 (TMTpro-18plex analysis of MM1.S cells treated with PROTACs), and PXD037179 (TMTpro-18plex analysis of NTERA-2 cells treated with PROTACs). Source data are provided with this paper.

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

## Acknowledgements

The authors thank C. Takahashi, Y. Horiuchi and T. Nakagawa for technical assistance and the Applied Protein Research Laboratory of Ehime University. We also thank Prof. K. Tohyama (Kawasaki Medical School) for providing the MDS-L cell line. We also thank Prof. F. Tokunaga and Dr D. Oikawa (Osaka Metropolitan University) for providing the B-cell lymphoma and DLBCL cell lines. This work was mainly supported by the Project for Cancer Research and Therapeutic Evolution (P-CREATE) from the Japan Agency for Medical Research and Development (AMED) under grant number JP21cm0106181h0006 (S.Yamanaka), Project for Promotion of Cancer Research and Therapeutic Evolution (P-PROMOTE) from AMED under grant number JP22cm0106181h0002 (S.Yamanaka), the Platform Project for Supporting Drug Discovery and Life Science Research (Basis for Supporting Innovative Drug Discovery and Life Science Research [BINDS]) from AMED under Grant Number 22ama121010j0001 (T.S.), a Grant-in-Aid for Scientific Research on Innovative Areas (21H00285 for S.Yamanaka, JP16H06579 for T.S. and JP19H04966 for H.K.) from the Japan Society for the Promotion of Science (JSPS). This work was also partially supported by JSPS KAKENHI (21K15076 for S.Y., JP19H03218 for T.S., and JP17H06112 for N.S.), Takeda Science Foundation, and Joint Usage and Joint Research Programs of the Institute of Advanced Medical Sciences, Tokushima University (T.S).

## Author contributions

S.Yamanaka, K.N. and Y.S. performed the biochemical, molecular, and cellular biology experiments. H.F., M.T. and T.M. performed ITC experiments and docking simulation. A.T., T.N., M.U. and N.S. synthesised and analysed the thalidomide derivatives and PROTACs. S.Yamanaka, Y.Y., S.Yoshida, and Y.I. analysed anti-proliferative effect of PROTACs. K.N. and H.K. performed TMT-based proteomics analyses. S.Yamanaka and T.S. analysed the data, designed the study, and wrote the paper, and all authors contributed to the manuscript.

## Competing interests

The authors declare no competing interests.
