## [Peer Review File · Nature Communications]

Lenalidomide derivatives and proteolysis-targeting chimeras for controlling neosubstrate degradationREVIEWER COMMENTS

Reviewer #1 (Remarks to the Author):

Over the last several years it has become well established that the ubiquitin proteasome can be co-opted through small molecule induced hijacking of E3 ligase enzymes for the purpose of redirecting substrate specificity to target disease-causing proteins. One of the two major types of these small molecule protein degraders are molecular glues. Molecular glues are a powerful type of degrader owing to their small size, bioavailability, and their proven track record of targeting new proteins that were previously considered undruggable. However, all the molecular glues to date have been discovered serendipitously such as Indisulam, Thalidomide and its analogs (collectively called IMiDs) and research efforts over the last few years have focused on identifying and understanding their target scope. In the case of IMiDs, they have both positive and negative phenotypes including targeting of IKFZ1/3 and CK1a to reduce heme cancer proliferation, but also targeting of SALL4 which is the likely cause of severe human birth defects when the drug is taken during the early weeks of pregnancy. Many groups are in the process of understanding the steep SAR surrounding these IMiD scaffolds and there is especially a lot of interest in understanding what is required to dial-in or dial-out some of these targets from a therapeutic perspective.

In this manuscript, the authors have explored modifications to the IMiD scaffold to identify the positions and modifications that confer neosubstrate selectivity. They have found that the 6- position on the glutaramide is essential for control of neosubstrate selectivity. The authors then used these new 6-position modified derivatives to make heterobifunctional degraders for BET proteins and showed that even with the addition of BET degradation, the molecules exhibited the same IMiD neosubstrate selectivity profile that the 6-position modified parental had.

The purpose of this study was to assess, demonstrate and understand tuning of the IMiD scaffold for the development of molecular glues and PROTAC molecules. IMiD molecules and derivatives hold huge therapeutic potential and studies surrounding their SAR and selectivity are of high importance to the field. There is also currently limited published data assessing the impact of minor chemical modifications to the IMiD scaffold – beyond SAR directed towards specific targets (eg, GSPT1, Helios). However, the manuscript is currently overstated claiming selectivity in molecules that have primarily been assessed focusing on assessment of a small number of the total reported targets. Addition of unbiased data would be important to back the selectivity claims made here and would be of much greater impact to the field.

Specific comments:

1. Would suggest including a mention of the clinical status of PROTACs when emphasizing the development of PROTACs worldwide on line 65.
2. The text is frequently incorrectly and under-referenced. Some examples include:
 - a. Ref's specifying some of the degradable targets of PROTACs (line 63 - currently only two reviews referenced and neither of these describe the diversity of targets or target binders)
 - b. Add Ref's for first small molecule PROTACs (line 62) – Winter et al., Science 2015 and Bondeson et al., Nat Chem Biol 2015)
 - c. Ref for CRBN binders (line 65)
 - d. Ref's for "several E3 binders have been developed". Binders have now extended beyond CRBN/IAP/VHL and it would be helpful to include references for these lesser known ligases – KEAP1, DCAF16, etc..
 - e. The references used for lines 69/70 do not support the text which says that neosubstrate degradation as a result of IMiD-based PROTACs. Please add appropriate references here, eg. Donovan et al., Cell 2020, Dobrovolsky et al., Blood 2019, etc..
 - i. Is there a published example of an IMiD-based PROTAC inducing degradation of PLZF?
 - f. Line 71/72 describes IMiD binding to the hydrophobic pocket. Neither of the two references for this statement describe this. Add Fischer et al., Nature 2014 and Chamberlain et al., Nat Struc Mol Biol 2014.

- g. There are more examples - please thoroughly check references throughout the text.
3. The statement: "These results suggest that 6-position-modification of lenalidomide is the optimal approach for selective thalidomide derivatives" is overstated and I would suggest diluting to specifically describe the selectivity that you are assessing in this case - eg, "is a viable approach for enhancing IKZF1 selectivity over SALL4". The literature and data show the SAR can be steep for these molecules and what works for enhancing selectivity of one target over another in one case may not work in the next case.
4. Figure 1b - Could the authors scale the y-axis of these plots for easier comparison.
5. Line 125: Could the authors clarify in the text that they are describing cellular ubiquitination.
6. Combine lines 147-149 with the following paragraph and the same with the section describing RUNX. It is easier for the reader to follow if the hypothesis statement is written together with the results of the experimental assessment.
7. I am unclear on the goal/purpose of paragraph 1 of section "PROTACs based on 6-modified lenalidomide". The beginning of the paragraph sounds like the authors are assessing the impact of altered linker on substrate specificity but the experimental test is on molecules that have altered target binding warheads, ligase binding warheads and linker chemistry/length making it impossible to derive any conclusion regarding which portion of the molecule is responsible for the differences in neosubstrate selectivity seen in the immunoblots. If the purpose is purely to set a baseline neosubstrate profile for each of the different selected PROTACs then simply stating this would be helpful.
8. In paragraph starting lines 189 the authors state that the STA-PDA assay "validated that 6-position modified lenalidomides with bulky molecules cannot interact with any neosubstrates in cells."
- a. The data is only shown for one of the two bulky-lenalidomides (F3CO-Le is missing - could this be included).
- b. Only four neosubstrates are tested - conclusion should be revised to make it clear
- c. The immunoblots look to show some very weak level of interaction that is above the DMSO baseline
9. Line 217: "PROTACs did not induce protein degradation in HEK293T and MDS-L cells" - please clarify which PROTACs and proteins this statement is referring to.
10. The authors discuss using antiproliferation assays as a measure of CRBN binding - a more direct read out for cellular CRBN binding would be using a cellular reporter assay such as that described in Nowak et al., Nat Chem Biol 2018. Were all these molecules assessed for cellular permeability? This would provide a more straightforward picture of the differences in cellular permeability and CRBN engagement.
11. Do the authors have any thoughts regarding the loss of binding affinity and selectivity for 6-Cl pomalidomide vs 6-Cl lenalidomide?
12. "In conclusion, more ideal thalidomide derivative-based PROTACs for TPD can be developed by combining the optimized linker and CRBN binders developed in this study" - I am unclear on which optimized linkers were developed in this study?
13. In determining selectivity and tuning out of IMiD neosubstrates (in proteomics and immunoblots), the authors have focused entirely on IKZF1/3 and SALL4 as the baseline for being defined as selective, however we know from Sievers et al., Science 2018 and Donovan et al., eLife 2018 that there are many other neosubstrates of IMiD molecules.
- a. The proteomics figures are currently selectively labeling relevant points - IKZF1/3, SALL4, BRD2/3/4 despite many other proteins having a higher FC and/or higher significance than these selected targets. What are these other proteins? Are some of these neosubstrates? Also, are the lenalidomide-based PROTACs hitting any additional targets that are not present in the pomalidomide-based PROTAC?
- b. Can the authors perform unbiased pulldowns (IP-MS) in cells or lysate to demonstrate binding selectivity beyond IKZF1/3 and SALL4 - supporting Extended data figure 2b,c.

Minor comments:

1. Line 63: Suggested edit: "theoretically many target proteins" to "theoretically target many proteins"

2. Line 76: Suggested edit: "selectivity of the neosubstrate" to "selectivity towards the neosubstrate"
3. Sentence on line 110-112 is not clear. Suggested edit: "Previous studies have reported that lenalidomide's ability to induce SALL4 degradation is the lowest amongst thalidomide derivatives currently in use"

Reviewer #2 (Remarks to the Author):

An interesting paper containing a lot of elegant experimental data. The idea of modifying Lenalidomide using the 6 position approach is solid with strong data supporting the claim made of new analogs that are more potent at targeting clinical conditions that Lenalidomide currently treats.

The authors appear to have identified some analogs that target substrates in clinical disease and avoid some substrates thought to be associated with teratogenesis, however it is likely there are more such targets than just SALL4 and PLZF involved, indeed the Authors mention these two targets have not been shown to reciprocate part or all the Thalidomide damage in mammalian models in-vivo. Perhaps I missed the point, why was p63 not looked for in these assays? Meis2 has also been linked to Thalidomide's teratogenic action. Other targets of Thalidomide thought to be involved in its teratogenic actions, and which have experimental evidence of being involved, include cell death induction, reactive oxygen species induction as well as inhibition of angiogenesis - there are studies that suggest Thalidomide's antiangiogenic action could be mediated via VEGF and other vascular markers. Were markers of such actions studied to indicate whether these were altered by the new analogs?

Anti-proliferative effects of the analogs were studied. Was cell death investigated? There is some evidence that fluorinated compounds can be cytotoxic.

Moreover, I would have liked to have seen some assays or evidence that the analogs are not likely to cause teratogenic effects in-vivo. The idea of making forms of a drug that keep the clinical relevance and reduce the side-effects is noble. Given the article is suggesting this is a novel way of making targeted forms of Lenalidomide and conclude they have done so, to target those involved in clinical disease, and hint at a reduced chance of targeting molecules linked to teratogenesis, could these compounds be used in an in-vivo setting to demonstrate this is the case? Zebrafish embryos have been used by several groups recently to study IMiDs actions and may prove useful in this case.

Reviewer #3 (Remarks to the Author):

In this manuscript, Yamanaka et al. develop structure-activity relationships for 6-substituted lenalidomide molecules, finding that, as compared to hydrogen at this position (unmodified lenalidomide), a fluorine atom induces greater recognition and degradation of therapeutic CRBN neo-substrates (IKZF1/3, CK1a) and increases the specificity for degradation of these substrates over teratogenic neo-substrates (SALL4, PLZF). Chlorine at this position shows the same specificity with weaker induction of binding and degradation. Larger substituents (Br, CF₃, OCF₃) maintain binding to CRBN but do not recruit typical CRBN neo-substrates, making PROTACS derived from these binders more specific degraders of intended PROTAC target. With respect to 6-fluoro lenalidomide, they make the case that, because of its enhanced recruitment of neo-substrates, it has greater antiproliferative effects than lenalidomide in MM and MDS cell lines sensitive to CRBN neo-substrate degradation. They also show that PROTACS derived from neo-substrate non-degrading CRBN binders are just as potent at degradation of the intended target as neo-substrate competent ones, and that they have similar viability effects in cell lines.

With regard to the SAR of the 6-halogenated lenalidomide series covered in the first two sections and figures 1 and 2, the data in aggregate sufficiently demonstrate the concluded SAR, however the individual data are qualitative rather than quantitative and the differences in the compounds are subtle in some cases. For instance, the comparison of Le and F-Le in Fig. 1b is purported to show that "6-fluoro lenalidomide (NE-005/F-Le) interacted with SALL4 at the same level as lenalidomide (Le) but interacted more strongly with IKZF1 than Le (Fig. 1a and b)." While the latter is certainly true, it's difficult to say from the curves that SALL4 is not also increased proportionally. Likewise, in fig 1c, despite an observable difference between them, it is unclear how significant the difference in SALL4 ubiquitination between Le and F-Le really is. While the cellular degradation data in figs. 1e and f do by eye appear to show greater IKZF and CK1 α degradation by F-Le, it is again difficult to quantitate or know exactly how significant this effect is. The strongest data in this section are the cellular ubiquitination data in fig. 1d, clearly showing better IKZF1 ubiquitination with F-Le than Le and the differences in SALL4 degradation in NTERA-2 cells in fig. 1g, demonstrating that F-Le is clearly less of a degrader than Le. Based on these data, which are easier to quantify by eye, and the trends of the other more subtle data, the purported SAR for this series is justified.

With respect to the cellular viability effects and mechanistic dissection of these effects illustrated in fig. 2, it is hard to determine without proper IC₅₀ determinations exactly how significant the differences between the compounds actually are, especially comparing F-Le to Le and Po. It appears from the data presented that at the concentrations tested, F-Le has more of an effect than Le, but these data need to be quantified in order to see the size of the effect and understand just how significant the SAR effect is. The PD readouts shown in fig 2c also seem to contradict the SAR effect with respect to Le vs F-Le, with Le having more of an effect than F-Le in most cases.

While the data provided here do generally demonstrate the purported SAR, which could be of general use, because of the lack of quantification of the SAR effect, it is hard to see how these findings alone advance the field. The significance of these findings could be made stronger by a much more quantitative treatment of the results to clearly demonstrate the effect. Importantly, it would be good to fully characterize the cooperativity within the SAR series, identifying just how much of an effect the different substituents have on the enhanced affinity of CRBN for the different neo-substrates. Additionally, quantification of the IC₅₀'s and D_{max} values for the different degraders in cellular assays would speak to the value of this SAR. Lastly, some structural characterization of the different ternary complexes to tie the differences in cooperativity and degradation to a physical mechanism would strengthen the significance of the findings.

In the subsequent section "PROTACs based on 6-modified lenalidomide," the data in Extended Data fig. 5 and 6 demonstrating that the 6-position modified lenalidomide derivatives all bind CRBN with relatively the same affinity but lose the ability to degrade neo-substrates as the size of the substituent increases is quite well exemplified, especially in ED fig. 5 b, c, and d, as well as ED fig. 6b. It would be good to use structural modeling to describe why this is the case. While the structural modeling may provide a very straightforward answer to this, it has not been previously exemplified in the literature.

The cellular degradation data in fig. 3b-d do clearly show that the bulkier 6-position substituent lenalidomide derived PROTAC molecules are as efficient at degrading BRD2/3/4 as the lenalidomide containing molecules, but show significantly less, if any, IKZF1 or SALL4 degradation, in line with the SAR. Based on the strength of the effect observable by eye and the fact that the data are being used to define an all-or-nothing event, no more detailed quantification of these results is required. The ability to target PROTAC substrates without degrading the canonical CRBN neo-substrates is of genuine interest to the field. It should be noted that this has been demonstrated previously (i.e. <https://doi.org/10.1002/anie.201901336>) albeit by modifying the PROTAC linker rather than by making substitutions to the IMiD itself.

ED fig. 8, used to support the statement "Lenalidomide and lenalidomide derivative-based PROTACs

more selectively induced the degradation of BET proteins than pomalidomide-based PROTACs," requires some annotation to allow the reader to determine whether some are more specific than others. As shown, the volcano plots all look extremely similar, with no clear way to tell which is degrading more non-specifically.

In the final section "Antiproliferative effect of PROTACs," the authors demonstrate that in cell lines for which canonical CRBN neo-substrate degradation has no viability effect, the different lenalidomide analog based PROTACS all degrade BRD proteins in the same dose range and have similar viability effects. While this is true, it is not commented on that the degraders are showing a hook effect wherein the lowest concentration tested (0.1uM) is actually showing the most robust degradation and at the highest concentration (10uM), the degradation is essentially blocked. While it is true at 100nM that the compounds all show the same 100% degradation and almost 0% viability, as the concentration increases to 1 and 10uM, the effects on viability must be coming, at least in part, from BRD inhibition rather than degradation, an effect which is not related to the CRBN binder. Thus, it is hard to compare the effects of degradation on viability at these high concentrations. In order to increase the significance of these findings, it would be good to do a fuller dose-response curve at lower concentrations below 100nM to assess the effects of BRD degradation on viability.

(ED fig 9 c, e, and f seem to be erroneously referred to in the text. These figures show a lack of viability effects for the non-PROTAC IMiD molecules in IMR32, NTERA-2, and HCT116 cell lines, but are referred to in the text as relating to PROTAC molecules.)

Reviewer #4 (Remarks to the Author):

In their manuscript entitled "Lenalidomide Derivative and PROTAC for Controlling Neosubstrate Degradation" by Yamanaka et al., the authors investigated the use of 6-position modifications of the thalidomide derivative lenalidomide as a strategy for selective targeted protein degradation (TPD) in the treatment of haematological cancers such as multiple myeloma (MM) and 5q myelodysplastic syndromes (5q MDS). The authors showed that 6-fluoro lenalidomide selectively degraded IKZF1, IKZF3, and CK1 α , which are involved in anti-haematological cancer activity, and had stronger antiproliferative effects on MM and 5q MDS cell lines than unmodified lenalidomide. The authors also demonstrated the use of proteolysis-targeting chimeras (PROTACs) incorporating 6-position-modified lenalidomide derivatives to selectively degrade BET proteins and inhibit proliferation in various cell lines including MM and neuroblastoma cell lines.

One potential concern is still the limited number of cell lines tested in the study. While the authors demonstrate the selective degradation of IKZF1, IKZF3, and CK1 α in MM and 5q MDS cell lines, it would be interesting to see if these effects are reproducible in a wider range of haematological cancer cell lines. Additionally, further investigation into the mechanisms behind the selective degradation of these neosubstrates and the potential for off-target effects would be beneficial.

Overall, this is a well-conducted and documented study (also regarding the proteomics experiments) with promising results that warrants further investigation into the use of 6-position-modified lenalidomide derivatives and PROTACs for selective TPD in the treatment of haematological cancers.

Point-by-Point Responses to the Reviewers' Critiques (NCOMMS-22-44598-T)

Reviewer #1

Over the last several years it has become well established that the ubiquitin proteasome can be co-opted through small molecule induced hijacking of E3 ligase enzymes for the purpose of redirecting substrate specificity to target disease-causing proteins. One of the two major types of these small molecule protein degraders are molecular glues. Molecular glues are a powerful type of degrader owing to their small size, bioavailability, and their proven track record of targeting new proteins that were previously considered undruggable. However, all the molecular glues to date have been discovered serendipitously such as Indisulam, Thalidomide and its analogs (collectively called IMiDs) and research efforts over the last few years have focused on identifying and understanding their target scope. In the case of IMiDs, they have both positive and negative phenotypes including targeting of IKFZ1/3 and CK1a to reduce heme cancer proliferation, but also targeting of SALL4 which is the likely cause of severe human birth defects when the drug is taken during the early weeks of pregnancy. Many groups are in the process of understanding the steep SAR surrounding these IMiD scaffolds and there is especially a lot of interest in understanding what is required to dial-in or dial-out some of these targets from a therapeutic perspective.

In this manuscript, the authors have explored modifications to the IMiD scaffold to identify the positions and modifications that confer neosubstrate selectivity. They have found that the 6- position on the glutaramide is essential for control of neosubstrate selectivity. The authors then used these new 6-position modified derivatives to make heterobifunctional degraders for BET proteins and showed that even with the addition of BET degradation, the molecules exhibited the same IMiD neosubstrate selectivity profile that the 6-position modified parental had.

The purpose of this study was to assess, demonstrate and understand tuning of the IMiD scaffold for the development of molecular glues and PROTAC molecules. IMiD molecules and derivatives hold huge therapeutic potential and studies surrounding their SAR and selectivity are of high importance to the field. There is also currently limited published data assessing the impact of minor chemical modifications to the IMiD scaffold – beyond SAR directed towards specific targets (eg, GSPT1, Helios). However, the manuscript is currently overstated claiming selectivity in molecules that have primarily been assessed focusing on assessment of a small number of the total

reported targets. Addition of unbiased data would be important to back the selectivity claims made here and would be of much greater impact to the field.

Response: We thank the reviewer for the kind comments and for raising important concerns about analysing unbiased neosubstrates.

Specific comments:

1. Would suggest including a mention of the clinical status of PROTACs when emphasizing the development of PROTACs worldwide on line 65.

Response: Thank you for your valuable suggestion. We described that several PROTACs are in clinical trials in the revised manuscript as follows: "Several PROTACs have been evaluated in clinical trials²²." (page 4, lines 63-64).

2. The text is frequently incorrectly and under-referenced. Some examples include:

a. Ref's specifying some of the degradable targets of PROTACs (line 63 - currently only two reviews referenced and neither of these describe the diversity of targets or target binders)

Response: Thank you for your important concerns about references. We have corrected references about diverse targets of PROTACs in the revised manuscript (page 4, line 61; page 4, line 63).

b. Add Ref's for first small molecule PROTACs (line 62) – Winter et al., Science 2015 and Bondeson et al., Nat Chem Biol 2015)

Response: We have added a reference for the first small molecule PROTAC (page 3, line 60, references 16 and 17).

c. Ref for CRBN binders (line 65)

Response: We have added reference for CRBN binders (page 4, line 65, reference 23).

d. Ref's for "several E3 binders have been developed". Binders have now extended beyond CRBN/IAP/VHL and it would be helpful to include references for these lesser known ligases – KEAP1, DCAF16, etc..

Response: We have added references for several E3 binders, such as VHL, cIAP, MDM2, and DCAF16 as follows: "Many E3 binders have been developed, including the CRBN binder^{16,23}, von Hippel–Lindau (VHL) binder^{23,24}, cellular inhibitor of apoptosis protein (cIAP) binder^{23,25}, mouse double minute 2 homolog (MDM2) binder^{23,26}, DDB1 and CUL4 associated factor 16 (DCAF16) binder^{23,27}, and DCAF11 binder²⁸." (page 4, lines 64-67)

e. The references used for lines 69/70 do not support the text which says that neosubstrate degradation as a result of IMiD-based PROTACs. Please add appropriate references here, eg. Donovan et al., Cell 2020, Dobrovolsky et al., Blood 2019, etc..

Response: We have added references for neosubstrate degradation by PROTACs (page 4, line 72, references 20 and 29).

i. Is there a published example of an IMiD-based PROTAC inducing degradation of PLZF?

Response: Because we have reported that PLZF was degraded by ARV-825 (Yamanaka et al., Nat. Commun. 13:183 (2022)), we have added the reference in the revised manuscript (reference 30).

f. Line 71/72 describes IMiD binding to the hydrophobic pocket. Neither of the two references for this statement describe this. Add Fischer et al., Nature 2014 and Chamberlain et al., Nat Struc Mol Biol 2014.

Response: We have added two references you mentioned in the revised manuscript (page 4, lines 73-74, references 31 and 32).

g. There are more examples - please thoroughly check references throughout the text.

Response: We thank you for your kind suggestions regarding references. We have thoroughly checked the references and corrected them in the revised manuscript.

3. The statement: “These results suggest that 6-position-modification of lenalidomide is the optimal approach for selective thalidomide derivatives” is overstated and I would suggest diluting to specifically describe the selectivity that you are assessing in this case – eg, “is a viable approach for enhancing IKZF1 selectivity over SALL4”. The literature and data show the SAR can be steep for these molecules and what works for enhancing selectivity of one target over another in one case may not work in the next case.

Response: We thank you for your important concerns that our statement is overstated. We have corrected it to our statement that the 6-position-modification is a viable approach for enhancing selectivity towards IKZF1 in the revised manuscript. We have added the following text: “These results suggest that 6-position-modification of lenalidomide is a viable approach for enhancing IKZF1 selectivity over SALL4.” (page 6, lines 109-111).

4. Figure 1b – Could the authors scale the y-axis of these plots for easier comparison.

Response: We thank you for pointing out the figure scales. In response, we repeated the AlphaScreen-based interaction assays using thalidomide, lenalidomide, pomalidomide, and lenalidomide derivatives and unified the Y-axis scale (Fig. 2b in the revised manuscript).

5. Line 125: Could the authors clarify in the text that they are describing cellular ubiquitination.

Response: We thank you for your important suggestion. We have clarified that the results (Fig. 2d in the revised manuscript) were of cellular ubiquitination assay. We have added revisions to the following text for clarity: “Consistent with results of the *in vitro* ubiquitination assay, the cellular polyubiquitination of SALL4 by F-Le was very weak (Fig. 2d). However, cellular polyubiquitination of IKZF1 was the strongest in HEK293T cells (Fig. 2d).” (page 7, lines 125-128).

6. Combine lines 147-149 with the following paragraph and the same with the section describing RUNX. It is easier for the reader to follow if the hypothesis statement is written together with the results of the experimental assessment.

Response: We thank you for your important suggestion about sentence structure. We have corrected the sentences in the revised manuscript (page 8, lines 152-157).

7. I am unclear on the goal/purpose of paragraph 1 of section “PROTACs based on 6-modified lenalidomide”. The beginning of the paragraph sounds like the authors are assessing the impact of altered linker on substrate specificity but the experimental test is on molecules that have altered target binding warheads, ligase binding warheads and linker chemistry/length making it impossible to derive any conclusion regarding which portion of the molecule is responsible for the differences in neosubstrate selectivity seen in the immunoblots. If the purpose is purely to set a baseline neosubstrate profile for each of the different selected PROTACs then simply stating this would be helpful.

Response: We thank you for your important suggestion about sentence structure. As you pointed out, our intention was to investigate whether the selected PROTACs induced the degradation of neosubstrates. In the revised manuscript, we simplified the sentences and made the following revisions: **“Degradation of target proteins and neosubstrates by PROTACs based on 6-modified lenalidomides**

IMiD-based PROTACs using various target binders have been developed^{19,20}, and PROTACs induced protein degradation of IMiD neosubstrates^{20,29}. IMiD-based PROTACs (Supplementary Fig. 7a) induced protein degradation of neosubstrates with different neosubstrate selectivities (Supplementary Fig. 7b and c). Therefore, we investigated whether 6-position-modified lenalidomides can be used as CRBN binders for PROTACs.” (page 15, lines 286-292).

8. In paragraph starting lines 189 the authors state that the STA-PDA assay “validated that 6-position modified lenalidomides with bulky molecules cannot interact with any neosubstrates in cells.”

a. The data is only shown for one of the two bulky-lenalidomides (F3CO-Le is missing – could this be included).

Response: We thank you for your important concerns. We have corrected the sentences in the revised manuscript (page 10, lines 191-193). In response to your comment, we experimentally addressed this concern by performing a pull-down assay using streptavidin beads. The results have been added to Fig. 4c in the revised manuscript. As shown in the results, two bulky-lenalidomides scarcely induced biotinylation of IKZF1, IKZF3, SALL4, and PLZF (Fig. 4c in the revised manuscript).

b. Only four neosubstrates are tested – conclusion should be revised to make it clear

Response: We thank you for your important concerns. In the revised manuscript, we made it clear that only four neosubstrates were examined (page 10, lines 191-193).

c. The immunoblots look to show some very weak level of interaction that is above the DMSO baseline

Response: We thank you for your suggestion. In response, we have corrected our sentence that two bulky-lenalidomides scarcely biotinylated the four neosubstrates in the revised manuscript. Our revision is as follows “The cellular binding abilities toward the neosubstrates were validated using streptavidin pull-down assay (STA-PDA) based on the proximity-dependent biotin identification (BioID) method, which was established in our previous studies^{30,38}, and it was shown that F₃C-Le and F₃CO-Le scarcely biotinylated SALL4, PLZF, IKZF1, and IKZF3 (Fig. 4c).” (page 10, lines 189-193).

9. Line 217: “PROTCs did not induce protein degradation in HEK293T and MDS-L cells” – please clarify which PROTACs and proteins this statement is referring to.

Response: We thank you for your important concerns. In the revised manuscript, we have specified the neosubstrates. Our revision is as follows: “Furthermore, F-P, Cl-P, and F₃C-P did not induce protein degradation of PLZF and CK1 α in HEK293T and MDS-L cells (Supplementary Fig. 9c and d).” (pages 17–18, lines 342-344).

10. The authors discuss using antiproliferation assays as a measure of CRBN binding – a more direct read out for cellular CRBN binding would be using a cellular reporter assay such as that described in Nowak et al., Nat Chem Biol 2018. Were all these molecules assessed for cellular permeability? This would provide a more straightforward picture of the differences in cellular permeability and CRBN engagement.

Response: We thank you for your constructive concerns about the CRBN binding ability of PROTACs. To experimentally address the concerns, we performed several experiments and added these analyses in the revised manuscript. First, we examined the CRBN binding ability of PROTACs based on 6-position-modified lenalidomide using biochemical assays. The CRBN binding ability of F₃C-P was only slightly lower than that of the other PROTACs (Fig. 6b). Then, we evaluated the drug-dependent formation of the ternary complex using biochemical assays. The hook effect of F₃C-P shifted at about a ten times higher dose than that of the other PROTACs (Fig. 6c). Finally, to quantitatively examine the degradation of a target protein, we generated HEK293T cells stably expressing BRD3-HiBit. F-P induced protein degradation at the same dose as that of Le-P, but BRD3-HiBit degradation by Cl-P and F₃C-P at low dose was weaker than that by Le-P and F-P (Fig. 6d). Therefore, we concluded that these lenalidomide derivatives can be used as CRBN binders, though optimization of the linker and cellular permeability may be required to achieve a higher degradation ability for target proteins. In the revised manuscript, we have discussed these results (pages 16–17, lines 316–335).

11. Do the authors have any thoughts regarding the loss of binding affinity and selectivity for 6-Cl pomalidomide vs 6-Cl lenalidomide?

Response: We thank you for your important concerns about the differences between 6-modified lenalidomide and 6-modified pomalidomide. In response, we have analysed docking models using crystal structures of IKZF1–CRBN, SALL4–CRBN, and CK1 α –CRBN complexes published in previous reports. 6-position substituents were oriented towards the β -hairpin structure of each neosubstrate (Supplementary Fig. 4b). Given that the binding ability of Cl-Le to CRBN was the same as that of F-Le (Supplementary Fig. 9c–e), we concluded that the different types and combinations of residues located around the 6-

position substituent among the three neosubstrates contribute to the neosubstrate selectivity. Therefore, we believe tiny differences between substituents on the phthalimide ring and the amino acid sequence of neosubstrate are critical for neosubstrate selectivity, though structural analyses are required to precisely understand the structural basis. In the revised manuscript, we have discussed the docking model (pages 11-12, lines 207–231).

12. “In conclusion, more ideal thalidomide derivative-based PROTACs for TPD can be developed by combining the optimized linker and CRBN binders developed in this study” – I am unclear on which optimized linkers were developed in this study?

Response: We thank you for your important concerns. We have clarified that this study did not optimize linkers in the revised manuscript. We have replaced the above text with the following: “Previous studies have shown that linkers and attachment sites between the E3 and target binders are critical for the degradation of target proteins and neosubstrates^{20,29,59}. Therefore, we believe that optimisations of the linker and attachment site may be required to develop more selective and effective PROTACs based on 6-position-modified lenalidomides.” (pages 23-24, lines 463–467).

13. In determining selectivity and tuning out of IMiD neosubstrates (in proteomics and immunoblots), the authors have focused entirely on IKZF1/3 and SALL4 as the baseline for being defined as selective, however we know from Sievers et al., Science 2018 and Donovan et al., eLife 2018 that there are many other neosubstrates of IMiD molecules.

a. The proteomics figures are currently selectively labeling relevant points – IKZF1/3, SALL4, BRD2/3/4 despite many other proteins having a higher FC and/or higher significance than these selected targets. What are these other proteins? Are some of these neosubstrates? Also, are the lenalidomide-based PROTACs hitting any additional targets that are not present in the pomalidomide-based PROTAC?

Response: We thank you for your important concerns about the neosubstrates of IMiDs. To address your concern, we have examined the degradation of neosubstrates reported in previous studies (Fig. 6i and Supplementary Fig. 10a–d). Pomalidomide-based PROTACs induced protein degradation of ZFP91, but lenalidomide-based PROTACs scarcely induced

ZFP91 degradation (Fig. 6i and Supplementary Fig. 10a–d). Alternatively, F-P and Cl-P degraded RAB28 though ARV-825 and L-P did not (Fig. 6i and Supplementary Fig. 10a–d). In addition, there were no selective neosubstrates that the lenalidomide-based PROTACs degraded (Fig. 6i and Supplementary Fig. 10a–d). Therefore, we concluded that PROTACs based on 6-position-modified lenalidomides reflected the neosubstrate selectivity of the CRBN binders. In the revised manuscript, we have discussed these results. (page 18, lines 348–364).

We have added the following text to explain further: “The MS analysis showed that protein levels of BET proteins were the same among all PROTACs (Fig. 6i and Supplementary Fig. 10a–b). Importantly, protein degradation levels of SALL4, IKZF1 and IKZF3 by F-P, Cl-P, and F₃C-P were lower than those by Po-P and Le-P (Fig. 6i and Supplementary Fig. 10a–b). Po-P also induced protein degradation of ZFP91 but Le-P, F-P, Cl-P, and F₃C-P did not (Fig. 6i and Supplementary Fig. 10a–b). Alternatively, F-P and Cl-P degraded RAB28 but Po-P, Le-P, and F₃C-P did not (Fig. 6i and Supplementary Fig. 10a–b). Notably, protein degradation levels of these neosubstrates by F₃C-P were the lowest among those by the PROTACs (Fig. 6i). These results suggest that the PROTACs retain neosubstrate selectivity of 6-position-modified lenalidomides.” (page 18, lines 350–358).

b. Can the authors perform unbiased pulldowns (IP-MS) in cells or lysate to demonstrate binding selectivity beyond IKZF1/3 and SALL4 – supporting Extended data figure 2b,c.

Response: We thank you for your important concerns about the neosubstrates of IMiDs. To experimentally address your concern, we performed TMT MS analyses and immunoblot analyses using 6-position-modified lenalidomides. F-Le and Cl-Le induced protein degradation of IKZF1, IKZF3, CK1 α , and RAB28 (Fig. 5a–f). However, the protein degradation ability of F-Le and Cl-Le for SALL4 and PLZF was lower than that of lenalidomide and pomalidomide (Fig. 5b). Importantly, two bulky-lenalidomides (F₃C-Le and F₃CO-Le) scarcely induced protein degradation of reported neosubstrates (Fig. 5a–f). These results support our argument that 6-position modification with small molecules enhances IKZF1 selectivity over SALL4, and bulky molecules cause a disruptive effect on neosubstrate degradation.

Minor comments:

1. Line 63: Suggested edit: “theoretically many target proteins” to “theoretically target many proteins”

Response: We have corrected the mistake in the revised manuscript (page 4, line 61).

2. Line 76: Suggested edit: “selectivity of the neosubstrate” to “selectivity towards the neosubstrate”

Response: We have changed the expression in the revised manuscript (page 4, line 78).

3. Sentence on line 110-112 is not clear. Suggested edit: “Previous studies have reported that lenalidomide’s ability to induce SALL4 degradation is the lowest amongst thalidomide derivatives currently in use”

Response: Our intention in the previous manuscript was that the binding and degradation ability of lenalidomide to SALL4 was lower than that of pomalidomide and thalidomide in previous studies. In the revised manuscript, we have modified the sentence for clarity.

Our revision is as follows: “Previous studies showed that both the binding and protein degradation abilities toward SALL4 of lenalidomide are lower than those of thalidomide and pomalidomide^{8,9}.” (page 6, lines 111–112).

Reviewer #2

An interesting paper containing a lot of elegant experimental data. The idea of modifying Lenalidomide using the 6 position approach is solid with strong data supporting the claim made of new analogs that are more potent at targeting clinical conditions that Lenalidomide currently treats.

Response: We thank the reviewer for the kind comments.

The authors appear to have identified some analogs that target substrates in clinical disease and avoid some substrates thought to be associated with teratogenesis, however it is likely there are more such targets than just SALL4 and PLZF involved, indeed the Authors mention these two targets have not been shown to reciprocate part or all the Thalidomide damage in mammalian models in-vivo. Perhaps I missed the point, why was p63 not looked for in these assays? Meis2 has also been linked to Thalidomide's teratogenic action. Other targets of Thalidomide thought to be involved in its teratogenic actions, and which have experimental evidence of being involved, include cell death induction, reactive oxygen species induction as well as inhibition of angiogenesis - there are studies that suggest Thalidomide's antiangiogenic action could be mediated via VEGF and other vascular markers.

Were markers of such actions studied to indicate whether these were altered by the new analogs?

Response: We thank you for your constructive suggestions and concerns. To address these concerns, we performed several experiments and added these analyses in the revised manuscript. Immunoblot analysis showed that all 6-position-modified lenalidomides did not degrade p63 (TP63 in Fig. 5d). However, all 6-position-modified lenalidomides increased the protein expression level of MEIS2 (Fig. 5e) because they bound with CRBN at the same level as did lenalidomide (Supplementary Fig. 9d and e). As the reviewer mentioned, we completely agree that many phenomena, such as cell death, generation of reactive oxygen species, and inhibition of angiogenesis, are involved in thalidomide-induced teratogenicity. However, given that thalidomide and its derivatives can bind with mouse Crbn but cannot induce neosubstrate degradation in mice, which do not show teratogenic phenotypes, we believe that neosubstrate degradation is the most critical, resulting in diverse phenomena

described above. Currently, only three neosubstrates, which are SALL4, PLZF, and p63, have been reported as neosubstrates directly involved in teratogenicity. Therefore, we investigated the protein degradation of the three neosubstrates in this study. In future studies, how the phenotype may be affected by 6-position-modified lenalidomides will be investigated.

Anti-proliferative effects of the analogs were studied. Was cell death investigated? There is some evidence that fluorinated compounds can be cytotoxic.

Response: We thank you for your important suggestion. To address your concern, we examined cell death induced by 6-position-modified lenalidomides. In HeLa cells, which are IMiD-non-sensitive cell lines, the combination of tumour necrosis factor- α (TNF α) and cycloheximide (CHX) induced typical apoptosis (Reviewer Fig. 1a and b). By contrast, all 6-position-modified lenalidomides did not induce apoptosis (Reviewer Fig. 1a and b).

Reviewer Fig. 1. Analyses of cytotoxicity of 6-position-modified lenalidomides.

a, Accelerated cell death by TNF α and CHX treatment. HeLa cells were treated with 20 ng/mL TNF α plus 100 μ g/mL CHX or 6-position-modified lenalidomides for 8 h, and the proteins were analysed using immunoblotting. **b**, Cytotoxicity of 6-position-modified lenalidomides. HeLa cells were treated with 20 ng/mL TNF α , 10 μ g/mL CHX, or 6-position-modified lenalidomides for 24 h, and cell viability was analysed using CellTiter-Glo assay.

Moreover, i would have liked to have seen some assays or evidence that the analogs are not likely to cause teratogenic effects in-vivo. tHe idea of making forms of a drug that keep the clinical relevance and reduce the side-effects is noble. Given the article is suggesting this is a novel way of making targeted forms of Lenalidomide and conclude they have done so, to target those involved in clinical disease, and hint at a reduced chance of targeting molecules linked to teratogenesis, could these compounds be used in an in-vivo setting to demonstrate this is the case? Zebrafish embryos have been used by several groups recently to study iMiDs actions and may prove useful in this case.

Response: We thank you for your constructive suggestions and important concerns. We completely agree with your comment that *in vivo* analyses are important to show the selectivity of the 6-position-modified lenalidomides. However, thalidomide and its derivatives are the typical drugs that show species specificity, and this issue prevents many researchers from analysing thalidomide-induced teratogenicity *in vivo*. Previous studies showed that rabbits and chickens showed teratogenic phenotypes. However, rabbit models are impractical because it requires large amounts of lenalidomide derivatives and individuals to evaluate teratogenic phenotypes. In chicken models, protein degradation of Plzf but not of Sall4 is induced, and therefore, it is difficult to precisely analyse teratogenic phenotypes. As the reviewer mentioned, several studies suggested that zebrafish also show thalidomide-induced teratogenicity. Therefore, we first examined whether zebrafish Crbn (DrCrbn) induced protein degradation of both human SALL4 (HsSALL4) and DrSall4 using CRBN-KO cells (Supplementary Fig. 5c and d). DrCrbn did not induce protein degradation of both HsSALL4 and DrSall4 (Supplementary Fig. 5c and d). By contrast, HsCRBN degraded both HsSALL4 and DrSall4 in the same experiment (Supplementary Fig. 5c and d), indicating that the zebrafish model is not suitable for the analysis of thalidomide teratogenicity. These results strongly suggest that a mammalian model using genome-edited mice is required to precisely analyse thalidomide-induced teratogenicity. Therefore, we focused on neosubstrate selectivity in this study, and protein degradation of broad neosubstrates was thoroughly examined in the revised manuscript (Fig. 5a–f). Our results and those of previous studies indicate that F-Le and Cl-Le are effective lenalidomide derivatives with reduced teratogenicity, though further *in vivo* experiments are required in future studies.

Reviewer #3

In this manuscript, Yamanaka et al. develop structure-activity relationships for 6-substituted lenalidomide molecules, finding that, as compared to hydrogen at this position (unmodified lenalidomide), a fluorine atom induces greater recognition and degradation of therapeutic CRBN neo-substrates (IKZF1/3, CK1a) and increases the specificity for degradation of these substrates over teratogenic neo-substrates (SALL4, PLZF). Chlorine at this position shows the same specificity with weaker induction of binding and degradation. Larger substituents (Br, CF₃, OCF₃) maintain binding to CRBN but do not recruit typical CRBN neosubstrates, making PROTACS derived from this binders more specific degraders of intended PROTAC target. With respect to 6-fluoro lenalidomide, they make the case that, because of its enhanced recruitment of neo-substrates, it has greater antiproliferative effects than lenalidomide in MM and MDS cell lines sensitive to CRBN neo-substrate degradation. They also show that PROTACs derived from neo-substrate non-degrading CRBN binders are just as potent at degradation of the intended target as neo-substrate competent ones, and that they have similar viability effects in cell lines.

Response: We thank you for the considerate comments.

With regard to the SAR of the 6-halogenated lenalidomide series covered in the first two sections and figures 1 and 2, the data in aggregate sufficiently demonstrate the concluded SAR, however the individual data are qualitative rather than quantitative and the differences in the compounds are subtle in some cases. For instance, the comparison of Le and F-Le in Fig. 1b is purported to show that “6-fluoro lenalidomide (NE-005/F-Le) interacted with SALL4 at the same level as lenalidomide (Le) but interacted more strongly with IKZF1 than Le (Fig. 1a and b).” While the latter is certainly true, it’s difficult to say from the curves that SALL4 is not also increased proportionally. Likewise, in fig 1c, despite an observable difference between them, it is unclear how significant the difference in SALL4 ubiquitination between Le and F-Le really is. While the cellular degradation data in figs. 1e and f do by eye appear to show greater IKZF and CK1 α degradation by F-Le, it is again difficult to quantitate or know exactly how significant this effect is. The strongest data in this section are the cellular ubiquitination data in fig. 1d, clearly showing better IKZF1 ubiquitination with F-Le

than Le and the differences in SALL4 degradation in NTERA-2 cells in fig. 1g, demonstrating that F-Le is clearly less of a degrader than Le. Based on these data, which are easier to quantify by eye, and the trends of the other more subtle data, the purported SAR for this series is justified.

Response: We thank you for your constructive suggestion to justify SARs of 6-position-modified lenalidomides. To address this concern, we quantitatively analysed protein degradation of IKZF1, SALL4, PLZF, and CK1 α using the HiBit system. First, we generated HEK293T cells stably expressing HiBit-tagged neosubstrates using lentivirus. Then, we examined the degradation ability of 6-position-modified lenalidomides for these neosubstrates. As shown in Fig. 4g, these results support the SARs of 6-position-modified lenalidomides. The quantitative values are shown in Fig. 4h, and we have described these results in the revised manuscript.

To clarify in further detail, we have added the following text to the manuscript: “Then, to quantitatively analyse the ability of 6-position-modified lenalidomides for neosubstrate degradation, we generated HEK293T cells stably expressing neosubstrate-HiBit using lentivirus. The luminescence signal of IKZF1-HiBit was markedly reduced by Po, Le, and F-Le in a dose-dependent manner (Fig. 4g) and slightly reduced by Cl-Le (Fig. 4g). In the case of CK1 α -HiBit, F-Le induced a more robust protein degradation of CK1 α -HiBit than did Le (Fig. 4g). Furthermore, the luminescence signals of SALL4-HiBit and PLZF-HiBit were markedly reduced by Po and slightly reduced by Le (Fig. 4g). Importantly, the degradation abilities of F-Le and Cl-Le were lower than that of Le, and reductions of the luminescence signals of the four neosubstrates-HiBit by Br-Le, F₃C-Le, and F₃CO-Le were scarcely observed (Fig. 4g). In addition, the degradation abilities of 6-position-modified lenalidomides for the four neosubstrates were quantified using DC₅₀ and D_{max} (Fig. 4h), supporting the SAR shown in Fig. 1–3 of the 6-position-modified lenalidomides.” (pages 10-11, lines 194-206).

With respect to the cellular viability effects and mechanistic dissection of these effects illustrated in fig. 2, it is hard to determine without proper IC₅₀ determinations exactly how significant the differences between the compounds actually are, especially comparing F-Le to Le and Po. It appears from the data presented that at the concentrations tested, F-Le has more of an effect than Le, but these data need to be

quantified in order to see the size of the effect and understand just how significant the SAR effect is. The PD readouts shown in fig 2c also seem to contradict the SAR effect with respect to Le vs F-Le, with Le having more of an effect than F-Le in most cases.

Response: We thank you for your constructive suggestion about quantitative experiments on cellular viability effects. To address this concern, we have performed cellular viability assays using lenalidomide derivatives in a wide range of concentrations. As shown in Fig. 3d and e, F-Le showed a stronger anti-proliferative effect than did Le. The quantitative values are shown in Fig. 3f as GI_{50} and GI_{max} , and we have described these results in the revised manuscript (page 9, lines 165–170). The results of Fig. 2c in the previous manuscript showed that the up-regulation of genes involved in differentiation into megakaryocytes was at the same level between Le and F-Le treatments (Fig. 3c in the revised manuscript). As described in the revised manuscript (page 8, lines 152–157), a previous study showed that this phenomenon was caused by the degradation of IKZF1. Because the IKZF1 degradation level by F-Le was slightly higher than that by Le (Fig. 3b), we believe that the results in Fig. 3c show the same level between F-Le and Le treatments. As described in the revised manuscript (page 9, lines 168–170), we concluded that the high effectiveness of F-Le for MDS-L results from the high degradation activity of F-Le for CK1 α .

While the data provided here do generally demonstrate the purported SAR, which could be of general use, because of the lack of quantification of the SAR effect, it is hard to see how these findings alone advance the field. The significance of these findings could be made stronger by a much more quantitative treatment of the results to clearly demonstrate the effect. Importantly, it would be good to fully characterize the cooperativity within the SAR series, identifying just how much of an effect the different substituents have on the enhanced affinity of CRBN for the different neo-substrates. Additionally, quantification of the IC₅₀'s and D_{max} values for the different degraders in cellular assays would speak to the value of this SAR. Lastly, some structural characterization of the different ternary complexes to tie the differences in cooperativity and degradation to a physical mechanism would strengthen the significance of the findings.

Response: We thank you for your important suggestions about SARs and structural characterization. As mentioned above, we performed quantitative experiments using the HiBit system (Fig. 4g in the revised manuscript). The DC₅₀ and Dmax values are shown in Fig. 4h, supporting the SARs of 6-position-modified lenalidomides. To understand the structural basis of 6-position-modified lenalidomides, we performed docking analyses using crystal structures of IKZF1–CRBN, SALL4–CRBN and CK1 α –CRBN complexes published in previous reports. 6-position substituents were oriented towards the β -hairpin structure of each neosubstrate (Supplementary Fig. 4b). Given that the binding ability of 6-position-modified lenalidomides for CRBN was at the same level as shown in Supplementary Fig. 9c–e, we concluded that the different types and combinations of residues located around the 6-position substituent among the three neosubstrates contribute to the neosubstrate selectivity (pages 21–22, lines 423–434 in the revised manuscript).

In the subsequent section “PROTACs based on 6-modified lenalidomide,” the data in Extended Data fig. 5 and 6 demonstrating that the 6-position modified lenalidomide derivatives all bind CRBN with relatively the same affinity but lose the ability to degrade neo-substrates as the size of the substituent increases is quite well exemplified, especially in ED fig. 5 b, c, and d, as well as ED fig. 6b. It would be good to use structural modeling to describe why this is the case. While the structural modeling may provide a very straightforward answer to this, it has not been previously exemplified in the literature.

Response: We thank you for your important suggestions about structural characterization. As mentioned above, we performed docking analyses. The docking model showed that all the 6-position substituents were located at the space bounded by the H353 residue of CRBN and some residues of the neosubstrates. However, it was suggested that the orientation change of these residues was required to allow the modifications with bulky substituents and may have a disruptive effect on the interaction between CRBN and neosubstrates. In the revised manuscript, we have discussed the docking model (pages 11–12, lines 217–231).

The cellular degradation data in fig. 3b-d do clearly show that the bulkier 6-position substituent lenalidomide derived PROTAC molecules are as efficient at degrading BRD2/3/4 as the lenalidomide containing molecules, but show significantly less, if any,

IKZF1 or SALL4 degradation, in line with the SAR. Based on the strength of the effect observable by eye and the fact that the data are being used to define an all-or-nothing event, no more detailed quantification of these results is required. The ability to target PROTAC substrates without degrading the canonical CRBN neo-substrates is of genuine interest to the field. It should be noted that this has been demonstrated previously (i.e. <https://doi.org/10.1002/anie.201901336>) albeit by modifying the PROTAC linker rather than by making substitutions to the IMiD itself.

Response: We thank you for your kind comments about the importance of neosubstrate degradation by PROTACs. As the reviewer mentioned, many studies reported that changes in linkers and attachment sites were critical for degradation of neosubstrates. In this study, we demonstrated that chemical modifications of thalidomide derivatives also affected degradation of neosubstrates. Because thalidomide derivatives may be generated by the metabolism of PROTACs, we believe that the improvement of thalidomide derivatives for their use as CRBN binders is also important. In the revised manuscript, we have discussed neosubstrate degradation by PROTACs, citing previous studies on PROTAC linkers and attachment sites (pages 23-24, lines 463–467).

ED fig. 8, used to support the statement “Lenalidomide and lenalidomide derivative-based PROTACs more selectively induced the degradation of BET proteins than pomalidomide-based PROTACs,” requires some annotation to allow the reader to determine whether some are more specific than others. As shown, the volcano plots all look extremely similar, with no clear way to tell which is degrading more non-specifically.

Response: We thank you for your important concerns regarding the proteomics results of PROTACs. In the revised manuscript, we have added an additional line on the X-axis to clarify whether lenalidomide and lenalidomide derivative-based PROTACs were more selective. In the results section, we have described the selectivity among PROTACs (page 18, lines 348-358).

In the final section “Antiproliferative effect of PROTACs,” the authors demonstrate that in cell lines for which canonical CRBN neo-substrate degradation has no viability

effect, the different lenalidomide analog based PROTACS all degrade BRD proteins in the same dose range and have similar viability effects. While this is true, it is not commented on that the degraders are showing a hook effect wherein the lowest concentration tested (0.1uM) is actually showing the most robust degradation and at the highest concentration (10uM), the degradation is essentially blocked. While it is true at 100nM that the compounds all show the same 100% degradation and almost 0% viability, as the concentration increases to 1 and 10uM, the effects on viability must be coming, at least in part, from BRD inhibition rather than degradation, an effect which is not related to the CRBN binder. Thus, it is hard to compare the effects of degradation on viability at these high concentrations. In order to increase the significance of these findings, it would be good to do a fuller dose-response curve at lower concentrations below 100nM to assess the effects of BRD degradation on viability.

Response: We thank you for your important concerns about the anti-proliferative effect of PROTACs. To experimentally address your concerns, we performed cellular viability assays using lenalidomide derivatives in a wide range of concentrations (Fig. 7a, c and e). Furthermore, we simultaneously examined the anti-proliferative effect of OTX-015 (Fig. 7a, c and e). Importantly, the G_{max} values showed that the PROTACs based on 6-position-modified lenalidomides had a higher anti-proliferative effect than did OTX-015 (Fig. 7f), indicating that the degradation of BET proteins contributed to the anti-proliferative effect of PROTACs. In the results section, we have discussed the anti-proliferative effect in the revised manuscript (pages 19-20, lines 361–393).

(ED fig 9 c, e, and f seem to be erroneously referred to in the text. These figures show a lack of viability effects for the non-PROTAC IMiD molecules in IMR32, NTERA-2, and HCT116 cell lines, but are referred to in the text as relating to PROTAC molecules.)

Response: We would like to apologize for the mistakes. In the revised manuscript, we have correctly referred to the results (Supplementary Fig. 11b, e, and f; page 19, lines 374–383; page 20, lines 385–386).

Reviewer #4

In their manuscript entitled "Lenalidomide Derivative and PROTAC for Controlling Neosubstrate Degradation" by Yamanaka et al., the authors investigated the use of 6-position modifications of the thalidomide derivative lenalidomide as a strategy for selective targeted protein degradation (TPD) in the treatment of haematological cancers such as multiple myeloma (MM) and 5q myelodysplastic syndromes (5q MDS). The authors showed that 6-fluoro lenalidomide selectively degraded IKZF1, IKZF3, and CK1 α , which are involved in anti-haematological cancer activity, and had stronger antiproliferative effects on MM and 5q MDS cell lines than unmodified lenalidomide. The authors also demonstrated the use of proteolysis-targeting chimeras (PROTACs) incorporating 6-position-modified lenalidomide derivatives to selectively degrade BET proteins and inhibit proliferation in various cell lines including MM and neuroblastoma cell lines.

One potential concern is still the limited number of cell lines tested in the study. While the authors demonstrate the selective degradation of IKZF1, IKZF3, and CK1 α in MM and 5q MDS cell lines, it would be interesting to see if these effects are reproducible in a wider range of haematological cancer cell lines. Additionally, further investigation into the mechanisms behind the selective degradation of these neosubstrates and the potential for off-target effects would be beneficial.

Overall, this is a well-conducted and documented study (also regarding the proteomics experiments) with promising results that warrants further investigation into the use of 6-position-modified lenalidomide derivatives and PROTACs for selective TPD in the treatment of haematological cancers.

Response: We thank you for your kind comments about the importance of this study. To experimentally address your concern, we added new results of anti-proliferative effects on six types of haematological cancer cell lines. F-Le and Cl-Le scarcely showed cytotoxicity to TK (B-cell non-Hodgkin lymphoma), BJAB (Burkitt lymphoma), Karpas-1106P (Primary mediastinal DLBCL), SU-DHL-4 (germinal centre B cell-like DLBCL), Jurkat (childhood T acute lymphoblastic leukaemia), and THP-1 (childhood acute monocytic leukaemia) cells (Supplementary Fig. 3a–f). Alternatively, pomalidomide showed cytotoxicity to TK and

BJAB cells (Supplementary Fig. 3a and b). However, previous studies suggested that lenalidomide showed an anti-proliferative effect on ABC-DLBCL but not on GCB-DLBCL. Furthermore, it was reported that lenalidomide was effective on adult T-cell leukaemia-lymphoma, follicular lymphoma, and marginal zone lymphoma. Therefore, although we focused on MM and 5q MDS, anti-proliferative effects on ABC-DLBCL and other lenalidomide-sensitive lymphomas should be investigated in future studies.

Furthermore, to experimentally address the your other concern, we thoroughly examined the protein degradation of neosubstrates using immunoblot analyses and quantitative proteomics (Fig. 5a–f). F-Le and Cl-Le induced protein degradation of IKZF1, IKZF3, CK1 α , and RAB28 (Fig. 5a–f). Alternatively, the protein degradation ability of F-Le and Cl-Le to SALL4 and PLZF was lower than that of lenalidomide and pomalidomide (Fig. 5b). These results support our argument that 6-position-modified lenalidomides showed an anti-proliferative effect on MM and 5q MDS via the degradation of IKZF1, IKZF3, and CK1 α . In the revised manuscript, we have discussed these results (pages 12-15, lines 235–284).

REVIEWERS' COMMENTS

Reviewer #1 (Remarks to the Author):

Over the last several years it has become well established that the ubiquitin proteasome can be co-opted through small molecule induced hijacking of E3 ligase enzymes for the purpose of redirecting substrate specificity to target disease-causing proteins. One of the two major types of these small molecule protein degraders are molecular glues. Molecular glues are a powerful type of degrader owing to their small size, bioavailability, and their proven track record of targeting new proteins that were previously considered undruggable. However, all the molecular glues to date have been discovered serendipitously such as Indisulam, Thalidomide and its analogs (collectively called IMiDs) and research efforts over the last few years have focused on identifying and understanding their target scope. In the case of IMiDs, they have both positive and negative phenotypes including targeting of IKZF1/3 and CK1a to reduce heme cancer proliferation, but also targeting of SALL4 which is the likely cause of severe human birth defects when the drug is taken during the early weeks of pregnancy. Many groups are in the process of understanding the steep SAR surrounding these IMiD scaffolds and there is especially a lot of interest in understanding what is required to dial-in or dial-out some of these targets from a therapeutic perspective.

In this manuscript, the authors have explored modifications to the IMiD scaffold to identify the positions and modifications that confer neosubstrate selectivity. They have found that the 6- position on the glutaramide is essential for control of neosubstrate selectivity. The authors then used these new 6-position modified derivatives to make heterobifunctional degraders for BET proteins and showed that even with the addition of BET degradation, the molecules exhibited the same IMiD neosubstrate selectivity profile that the 6-position modified parental had.

The purpose of this study was to assess, demonstrate and understand tuning of the IMiD scaffold for the development of molecular glues and PROTAC molecules. IMiD molecules and derivatives hold huge therapeutic potential and studies surrounding their SAR and selectivity are of high importance to the field. There is also currently limited published data assessing the impact of minor chemical modifications to the IMiD scaffold – beyond SAR directed towards specific targets (eg, GSPT1, Helios).

The manuscript is much more clear and easier to read and the authors have sufficiently addressed the reviewers previous comments.

- 1) The new text provides some analysis of docking poses and suggests differences in residues near the 6'-position in neosubstrates may contribute to the selectivity seen. Are there any examples of other 6'-modified IMiDs that have been published that follow similar trends in selectivity?
- 2) There is a statement describing that the 6-position-modified lenalidomides increased the expression level of MEIS2 similarly to Le and Po treatment (Fig. 5e), however the blot does not show very clear increased expression of MEIS2 compared to DMSO - can the authors quantify the changes here?

Minor things for clarity:

1) Page 11, 217:

"All the 6-position substituents were located at the space bounded by the H353 residue of CRBN and some residues of the neosubstrates (Q146 and N148 for IKZF1, V411 and S413 for SALL4, and K18, I35 and I37 for CK1a)." -

Rephrase to: "All the 6-position substituents were located in the space bound by the H353 residue of CRBN and some residues of the neosubstrates (Q146 and N148 for IKZF1, V411 and S413 for SALL4, and K18, I35 and I37 for CK1a)."

2) Page 12, 228:

"whereas no such residue was not found in SALL4" -

rephrase to "whereas no such residue was found in SALL4"

3) Page 13, 261:

"Immunoblot analyses showed that human CRBN(HsCRBN) induced protein degradation of SALL4 but

not of zebrafish Crbn (DrCrbn)(Supplementary Fig. 5c and d), indicating that zebrafish model cannot be used to precisely evaluate thalidomide-induced teratogenicity."

Rephrase to: "Immunoblot analyses showed that human CRBN(HsCRBN) can induce degradation of SALL4, but zebrafish Crbn (DrCrbn) can not(Supplementary Fig. 5c and d), indicating that a wild type zebrafish model cannot be used to precisely evaluate thalidomide-induced teratogenicity."

Reviewer #2 (Remarks to the Author):

I congratulate the Authors on addressing all the comments and queries in the revised manuscript, and including the resulting new data in the manuscript.

In this revised manuscript the authors make several points that Zebrafish are not ideal to use to study thalidomide or are controversial to be used to study limb teratogenesis. However zebrafish were used to confirm the role of Cereblon itself in thalidomide teratogenesis (Ito et al 2010) and very recently a paper was published using zebrafish fin to identify and then model gene function that underpin human limb malformations (Truong et al., 2023.). So zebrafish do have a use, but is likely to be context dependent.

Reviewer #3 (Remarks to the Author):

I want to thank the authors of the manuscript for their attention to detail in thoroughly addressing my concerns about the original manuscript, namely that quantitative SAR was required to demonstrate the full significance of the findings. These new data include full or fuller dose-responses in the alpha screen, cell viability, and cellular degradation assays. I am especially sensitive to the fact that developing half a dozen HiBit lines is no simple task, but the data are greatly beneficial to the conclusions. The structural modeling of the compounds binding to the CRBN complexes nicely provides a possible explanation for the SAR. Overall, my concerns have been met and I would recommend publication in Nature Communications.

There are a few places where I would recommend small revisions.

Line 126-7: "polyubiquitination of SALL4 by F-Le was / very weak" - I would drop the "very"

Line 131: "but the induction of SALL4 and PLZF degradation by F-Le was weaker" - I don't think this is true for PLZF

Line 151: "also reduced the protein expression levels of IRF4 and Myc" - while technically true, the effect for IRF4 is very subtle.

Reviewer #4 (Remarks to the Author):

In the revised version the authors have adequately addressed the criticism raised by me, and also added new experimental data.

Point-by-Point Responses to the Reviewers' Critiques (NCOMMS-22-44598-A)

Reviewer #1

Over the last several years it has become well established that the ubiquitin proteasome can be co-opted through small molecule induced hijacking of E3 ligase enzymes for the purpose of redirecting substrate specificity to target disease-causing proteins. One of the two major types of these small molecule protein degraders are molecular glues. Molecular glues are a powerful type of degrader owing to their small size, bioavailability, and their proven track record of targeting new proteins that were previously considered undruggable. However, all the molecular glues to date have been discovered serendipitously such as Indisulam, Thalidomide and its analogs (collectively called IMiDs) and research efforts over the last few years have focused on identifying and understanding their target scope. In the case of IMiDs, they have both positive and negative phenotypes including targeting of IKFZ1/3 and CK1a to reduce heme cancer proliferation, but also targeting of SALL4 which is the likely cause of severe human birth defects when the drug is taken during the early weeks of pregnancy. Many groups are in the process of understanding the steep SAR surrounding these IMiD scaffolds and there is especially a lot of interest in understanding what is required to dial-in or dial-out some of these targets from a therapeutic perspective.

In this manuscript, the authors have explored modifications to the IMiD scaffold to identify the positions and modifications that confer neosubstrate selectivity. They have found that the 6- position on the glutaramide is essential for control of neosubstrate selectivity. The authors then used these new 6-position modified derivatives to make heterobifunctional degraders for BET proteins and showed that even with the addition of BET degradation, the molecules exhibited the same IMiD neosubstrate selectivity profile that the 6-position modified parental had.

The purpose of this study was to assess, demonstrate and understand tuning of the IMiD scaffold for the development of molecular glues and PROTAC molecules. IMiD molecules and derivatives hold huge therapeutic potential and studies surrounding their SAR and selectivity are of high importance to the field. There is also currently limited published data assessing the impact of minor chemical modifications to the IMiD scaffold – beyond SAR directed towards specific targets (eg, GSPT1, Helios).

The manuscript is much more clear and easier to read and the authors have sufficiently addressed the reviewers previous comments.

Response: We thank you for your kind comments and valuable suggestions on our revised manuscript.

1) The new text provides some analysis of docking poses and suggests differences in residues near the 6'-position in neosubstrates may contribute to the selectivity seen. Are there any examples of other 6'-modified IMiDs that have been published that follow similar trends in selectivity?

Response: We thank the reviewer for their important comments about previous studies on 6'-modified IMiDs. Previous studies revealed that CC-885 (5'-modified derivatives) induced the protein degradation of GSPT1, although lenalidomide and pomalidomide cannot degrade GSPT1 (Matyskiela et al., Nature 535, 252–257 (2016)). Furthermore, it was reported that CC-220 (4'-modified derivatives) induced strong degradation of IKZF1 and IKZF3 at a lower dose than lenalidomide and pomalidomide (Matyskiela, M.E. et al., J. Med. Chem. 61, 535–542 (2018)). However, to our knowledge, degradation with 6'-modified derivatives has not been reported. This study showed that inserting a second substituent to the 6'-position of lenalidomide alters neosubstrate selectivity. Therefore, we believe that 6'-position modified CC-220 and CC-92480 may be more selective and stronger thalidomide derivatives for IKZF1 and IKZF3 (page 23, lines 429-434), although further analyses are required.

2) There is a statement describing that the 6-position-modified lenalidomides increased the expression level of MEIS2 similarly to Le and Po treatment (Fig. 5e), however the blot does not show very clear increased expression of MEIS2 compared to DMSO - can the authors quantify the changes here?

Response: We thank the reviewer for constructive concerns about the MEIS2 expression level. To address the reviewer's concerns, we repeated the experiments (n=3), quantified the band intensity of MEIS2, and added the graph to Figure 5e in the revised manuscript. The 6'-position-modified lenalidomides slightly increased the expression level of MEIS2, similar to lenalidomide and pomalidomide (Fig. 5e). In the revised manuscript, we specified that the changes were subtle (page 15, lines 282-284).

Minor things for clarity:

1) Page 11, 217:

"All the 6-position substituents were located at the space bounded by the H353 residue of CRBN and some residues of the neosubstrates (Q146 and N148 for IKZF1, V411 and S413 for SALL4, and K18, I35 and I37 for CK1 α)." -

Rephrase to: "All the 6-position substituents were located in the space bound by the H353 residue of CRBN and some residues of the neosubstrates (Q146 and N148 for IKZF1, V411 and S413 for SALL4, and K18, I35 and I37 for CK1 α)."

Response: We have rephrased the sentence in the revised manuscript (page 13, lines 230-233).

2) Page 12, 228:

"whereas no such residue was not found in SALL4" -

rephrase to "whereas no such residue was found in SALL4"

Response: We have rephrased the sentence in the revised manuscript (page 13, lines 240-241).

3) Page 13, 261:

"Immunoblot analyses showed that human CRBN(HsCRBN) induced protein degradation of SALL4 but not of zebrafish Crbn (DrCrbn)(Supplementary Fig. 5c and d), indicating that zebrafish model cannot be used to precisely evaluate thalidomide-induced teratogenicity."

Rephrase to: "Immunoblot analyses showed that human CRBN(HsCRBN) can induce degradation of SALL4, but zebrafish Crbn (DrCrbn) can not(Supplementary Fig. 5c and d), indicating that a wild type zebrafish model cannot be used to precisely evaluate thalidomide-induced teratogenicity."

Response: We have rephrased the sentence in the revised manuscript (page 15, lines 274-277).

Reviewer #2

I congratulate the Authors on addressing all the comments and queries in the revised manuscript, and including the resulting new data in the manuscript.

In this revised manuscript the authors make several points that Zebrafish are not ideal to use to study thalidomide or are controversial to be used to study limb teratogenesis. However zebrafish were used to confirm the role of Cereblon itself in thalidomide teratogenesis (Ito et al 2010) and very recently a paper was published using zebrafish fin to identify and then model gene function that underpin human limb malformations (Truong et al., 2023.). So zebrafish do have a use, but is likely to be context dependent.

Response: Thank you for the kind comments and valuable suggestions regarding the zebrafish model. We completely agree with your comment that the zebrafish model is useful for evaluating human limb teratogenicity. However, because zebrafish Crbn cannot induce the degradation of neosubstrates, such as Sall4, we concluded that it is difficult to precisely evaluate thalidomide derivative-induced teratogenicity using the zebrafish model. In the revised manuscript, we have described the zebrafish model to precisely evaluate thalidomide derivative-induced teratogenicity as follows: "p63 is a neosubstrate involved in thalidomide teratogenicity in zebrafish⁴⁸, but the use of the zebrafish model to analyse limb teratogenicity induced by thalidomide derivatives is controversial⁴⁹. Importantly, the 6-position-modified lenalidomides did not induce protein degradation of p63 (TP63) (Fig. 5d). Immunoblot analyses showed that human CRBN (HsCRBN) can induce the degradation of SALL4, but zebrafish Crbn (DrCrbn) cannot (Supplementary Fig. 5c and d), indicating that a wild type zebrafish model cannot be used to precisely evaluate thalidomide-induced teratogenicity.

Reviewer #3

I want to thank the authors of the manuscript for their attention to detail in thoroughly addressing my concerns about the original manuscript, namely that quantitative SAR was required to demonstrate the full significance of the findings. These new data include full or fuller dose-responses in the alpha screen, cell viability, and cellular degradation assays. I am especially sensitive to the fact that developing half a dozen HiBit lines is no simple task, but the data are greatly beneficial to the conclusions. The structural modeling of the compounds binding to the CRBN complexes nicely provides a possible explanation for the SAR. Overall, my concerns have been met and I would recommend publication in Nature Communications.

There are a few places where I would recommend small revisions.

Response: We thank you for your kind comments on our revised manuscript.

Line 126-7: "polyubiquitination of SALL4 by F-Le was / very weak" - I would drop the "very"

Response: We have removed the "very" in the revised manuscript (page 8, line 134).

Line 131: "but the induction of SALL4 and PLZF degradation by F-Le was weaker" - I don't think this is true for PLZF

Response: We have corrected the mistake in the revised manuscript (page 9, line 142).

Line 151: "also reduced the protein expression levels of IRF4 and Myc" - while technically true, the effect for IRF4 is very subtle.

Response: We thank the reviewer for the important comment. In the revised manuscript, we specified that the reduction of IRF4 is subtle (page 10, lines 162-163).

Reviewer #4

In the revised version the authors have adequately addressed the criticism raised by me, and also added new experimental data.

Response: We thank you for your kind comments on our revised manuscript.